# ACCELERATING ATTENTION WITH BASIS DECOMPOSITION

## ABSTRACT

Attention is a core operation in large language models (LLMs) and vision-language models (VLMs). We present BD Attention (**BDA**), the first *lossless algorithmic reformulation* of attention. BDA is enabled by a simple matrix identity from Basis Decomposition (**BD**), which restructures multi-head projections into a compact form while preserving exact outputs. Unlike I/O-aware system optimizations such as FlashAttention, BDA provides a mathematically guaranteed acceleration that is architecture-agnostic. On DeepSeek-V2-Lite (16B, FP16), BDA requires only **4s** of offline preparation **with no retraining required** and, on modern GPUs, achieves **32% faster** key/value projections and **25% smaller** weights, while increasing end-to-end perplexity (PPL) by just **0.02%** (FP16) or **0.0004%** (FP32)—a negligible effect on model performance. These results position BDA as the first theoretically exact method for lossless attention acceleration that is complementary to existing engineering-level optimizations. Our code is available at `https://anonymous.4open.science/r/Basis-decomp-57B8`.

## 1 INTRODUCTION

Attention (Vaswani et al., 2017) has emerged as the fundamental building block in large language models (LLMs) and vision-language models (VLMs), enabling them to scale effectively across diverse tasks. However, the cost of multi-head attention (MHA) in both computation and memory makes it a major efficiency bottleneck. Existing acceleration techniques can be broadly divided into two categories. On the one hand, *lossless system-level optimizations*, such as FlashAttention (Dao et al., 2022), improve I/O efficiency by reordering memory access and fusing kernels, but their gains are hardware-specific and do not reduce the number of arithmetic operations or parameters. On the other hand, *approximate algorithmic approaches*, including linear attention (Katharopoulos et al., 2020; Choromanski et al., 2021), sparse attention (Child et al., 2019; Beltagy et al., 2020), pruning (Ma et al., 2023; Frantar & Alistarh, 2023), and quantization (Frantar et al., 2023; Lin et al., 2024c), reduce complexity or storage but inevitably trade off accuracy and often require retraining or calibration.

In this work, we take a different path and propose **BD Attention (BDA)**, the first *lossless algorithmic reformulation* of attention. BDA is enabled by a simple yet general matrix identity from **Basis Decomposition (BD)** to restructure MHA projections into a compact form, thereby reducing parameters and arithmetic operations while preserving exact outputs. This algorithmic perspective complements existing I/O-aware optimizations.

We validate BDA across multiple scenarios. On DeepSeek-V2-Lite (16B, FP16), BDA requires only **4s** of offline preparation with no retraining and achieves **32% faster** key/value projections and **25% smaller** weights on modern GPUs, with end-to-end perplexity (PPL) change at the negligible level of **0.02%**. Beyond inference, training experiments show that BDA achieves BLEU scores comparable to MHA without any hyperparameter adjustment. Furthermore, when applied on top of low-rank pruned models, BD reduces memory by **16.5%** and improves throughput by **17.2%**, demonstrating its compatibility with existing compression techniques.

Overall, BDA establishes a new algorithmic foundation for lossless attention acceleration, uniting theoretical guarantees with practical speedups on real hardware.

We highlight two main contributions of this study:

Table 1: **Comparison of acceleration techniques for Transformers.**

| Method | Lossless | (Re)Train | CPU Speedup | GPU Speedup | # Params |
|---|---|---|---|---|---|
| Flash Attention | ✓ | ✗ | ✗ | ✓ | Same |
| Linear / Sparse Attention | ✗ | ✓ | ✓ | ✓ | Same |
| Pruning / Quantization | ✗ | Partial | ✓ | ✓ | Reduce |
| BD Attention | ✓ | ✗ | ✓ | ✓ | Reduce |

- **Theoretical foundation.** We propose **Basis Decomposition (BD)**, a general matrix decomposition that guarantees hardware-friendly reconstruction with probability 1 under assumptions that are naturally satisfied by neural network weight matrices (Theorem 3.1).

- **Practical acceleration.** We apply BD to attention and low-rank layers, showing that it reduces parameters and arithmetic operations without affecting model quality. We further design an efficient Triton kernel that fuses BDA's operator steps into a single GPU kernel, achieving near-theoretical speedups in practice.

## 2 RELATED WORK

Research on accelerating attention broadly falls into two categories: (i) *lossless* system-level optimizations that preserve exact outputs, and (ii) *approximate* algorithm that trade accuracy for efficiency. Table 1 summarizes representative approaches.

**Lossless methods.** FlashAttention and its successors (Dao et al., 2022; Dao, 2023; Shah et al., 2024; Kwon et al., 2023) reduce memory traffic by fusing attention kernels and tiling reads/writes to on-chip SRAM, achieving substantial GPU speedups without altering outputs. These approaches are *lossless* but depend heavily on GPU memory hierarchies and I/O architectures.

**Approximate methods.** In contrast, many lines of work accelerate attention or reduce memory by introducing approximations.

- **Linear-attention** replaces the quadratic softmax kernel with linear-time kernel approximations (Wang et al., 2020; Katharopoulos et al., 2020; Choromanski et al., 2021).

- **Sparse-attention** imposes predefined or learned sparsity patterns, e.g., Sparse Transformers (Child et al., 2019), Longformer (Beltagy et al., 2020), Reformer (Kitaev et al., 2020).

- **Pruning** removes parameters based on importance criteria.
  - **Unstructured pruning** targets individual weights (Frantar & Alistarh, 2023; Sun et al., 2024; Zhang et al., 2024), effective in reducing parameters but hard to accelerate on GPUs.
  - **Semi-structured pruning** (e.g., $N : M$ patterns) (Mishra et al., 2021) enables GPU acceleration on NVIDIA's Ampere GPUs but restricts density flexibility.
  - **Structured pruning** removes entire channels/heads (Ma et al., 2023; van der Ouderaa et al., 2024; Ashkboos et al., 2024), more deployment-friendly but often with higher accuracy loss.
  - **Low-rank pruning** approximates weight matrices by decomposing them into lower-rank components (Yuan et al., 2023; Wang et al., 2024; Zhao et al., 2025), reducing both parameters and compute. While hardware-friendly, it introduces approximation error unless the true rank is low.

- **Quantization** compresses parameters/activations into low-precision formats (Frantar et al., 2023; Lin et al., 2024c; Xiao et al., 2023; Lin et al., 2024b), achieving strong memory savings but inevitably introducing errors at low bit-widths.

These methods can deliver large efficiency gains but are inherently *not lossless*, and usually require careful finetuning to mitigate quality drops.

# 3 METHOD

Figure 1: Illustration of BD Attention (**BDA**) using the QK projection as an example (VO is analogous). BDA consists of two stages: (a) **BD Attention Preparation** (Algorithm 3), performed offline *once* during model deployment, where the projection matrices are transformed via Basis Decomposition; (b) **BD Attention Inference** (Algorithm 2) saves $d_h/d$ in both parameters and computation, while preserving exact outputs.

## 3.1 BASIS DECOMPOSITION

Given two rectangular matrices $\mathbf{U} \in \mathbb{R}^{m \times r}$ and $\mathbf{V}^\top \in \mathbb{R}^{r \times n}$, where $r < \min(m, n)$, their product matrix $\mathbf{W} = \mathbf{U}\mathbf{V}^\top$ has rank at most $r$. Our goal is to develop a general decomposition method that expresses $\mathbf{W}$ as $\mathbf{W} = f(\mathbf{M}_1, \dots, \mathbf{M}_k)$, where the function $f(\mathbf{M}_1, \dots, \mathbf{M}_k)$ has lower computational cost than the original multiplication $\mathbf{U}\mathbf{V}^\top$.

We assume $\mathbf{W}$ has rank $r$ without loss of generality. In this case, there exist exactly $r$ linearly independent rows in $\mathbf{W}$. We define a matrix $\mathbf{B} = [\mathbf{b}_1, \dots, \mathbf{b}_r]^\top \in \mathbb{R}^{r \times n}$, where each $\mathbf{b}_j$ is one of these linearly independent rows. Thus, $\mathbf{B}$ forms a basis of the row space of $\mathbf{W}$. For any row $\mathbf{w}_i \in \mathbf{W} \setminus \mathbf{B}$, where $\mathbf{w}_i \in \mathbb{R}^n$, it can be written as a linear combination of the basis vectors in $\mathbf{B}$:

$$\mathbf{w}_i = \sum_{j=1}^{r} c_{ij} \mathbf{b}_j, \tag{1}$$

which follows directly from the fact that the row space of $\mathbf{W}$ is spanned by the $r$ basis vectors in $\mathbf{B}$. Collecting all such coefficients $c_{ij}$ forms a coefficient matrix $\mathbf{C} \in \mathbb{R}^{(m-r) \times r}$, where each row contains the weights for reconstructing a non-basis row of $\mathbf{W}$ as a linear combination of the basis vectors in $\mathbf{B}$. With $\mathbf{B}$ and $\mathbf{C}$, the original matrix $\mathbf{W}$ can be fully reconstructed.

This forms an alternative representation of the low-rank matrix $\mathbf{W}$, distinct from the traditional low-rank multiplication $\mathbf{U}\mathbf{V}^\top$. We refer to this representation as **Basis Decomposition (BD)**, encompassing both the decomposition $\mathbf{W} \rightarrow (\mathbf{B}, \mathbf{C})$ and its reconstruction $(\mathbf{B}, \mathbf{C}) \rightarrow \mathbf{W}$. We use the term *BD* for general reference, and specify *BD decomposition/reconstruction* only when emphasizing the decomposition/reconstruction process.

**Memory cost of BD.** BD stores two matrices: the basis matrix $\mathbf{B} \in \mathbb{R}^{r \times n}$ and the coefficient matrix $\mathbf{C} \in \mathbb{R}^{(m-r) \times r}$. The total memory cost is $r(m + n - r)$, which is strictly smaller than the full matrix size $mn$ and low rank matrices size $r(m + n)$ for any $r < \min(m, n)$. In contrast, the traditional low-rank representation $\mathbf{W} = \mathbf{UV}^\top$ requires $r(m + n)$ parameters and is only more compact than the full matrix when $r < \frac{mn}{m+n}$.

**Computational cost of BD reconstruction.** Reconstructing $\mathbf{W}$ from BD involves computing $\mathbf{CB}$ and inserting the $r$ basis rows. This requires $2r(m - r)n$ floating-point operations (FLOPs). In comparison, the traditional reconstruction $\mathbf{UV}^\top$ costs $2rmn$ FLOPs. Therefore, BD reconstruction is computationally more efficient for any $r < \min(m, n)$.

## 3.2 Selection of Basis

**Theorem 3.1** (Almost Sure Full Rank of Random Matrices). *Let $\mathbf{W}$ be an $r \times r$ real random matrix. Suppose the entries of $\mathbf{W}$ are drawn from a probability measure $\mu$ on $\mathbb{R}^{r^2}$ that is absolutely continuous with respect to the Lebesgue measure $\lambda$. Then $W$ has full rank ($\mathrm{rank}(W) = r$) with probability 1.*

Theorem 3.1 (proof in Appendix A) states that if a matrix $\mathbf{M} \in \mathbb{R}^{r \times r}$ has entries drawn from a distribution that is absolutely continuous with respect to the Lebesgue measure, then $\mathbf{M}$ is full rank with probability 1. In practical terms, this applies to any matrix with random noise of arbitrary (non-degenerate) scale.

Therefore, when $\mathbf{W}$ is generated as a product of noised matrices $\mathbf{UV}^\top$, we can safely assume in practice that any selection of $r$ rows from $\mathbf{W}$ forms a matrix $\mathbf{B} \in \mathbb{R}^{r \times n}$ that is full-rank. This situation is common in practice—for example, when $\mathbf{U}$ and $\mathbf{V}$ are weight matrices learned via stochastic gradient descent (SGD) in neural networks, where random initialization and noisy updates introduce sufficient perturbations. As a result, any $r \times r$ submatrix of $\mathbf{B}$, formed by choosing $r$ columns, is full rank with probability 1, resulting that the rows of $\mathbf{B}$ are linearly independent. This allows us to **freely choose** any $r$ rows (or $r$ columns) as a valid basis for BD reconstruction, without requiring explicit rank analysis or rotating.

Notably, PIFA (Zhao et al., 2025) can be regarded as a special case of BD: it selects basis rows via QR factorization with column pivoting, approximating the most numerically independent directions (Businger & Golub, 1971). While this is useful in rare rank-deficient or ill-conditioned settings, such guarantees are unnecessary in typical neural networks where noise ensures full rank. Hence, BD offers a more flexible and efficient basis selection strategy.

In particular, we find that choosing contiguous rows/columns, such as the first-$r$ or last-$r$ rows/columns, offers significant efficiency advantages. It minimizes the I/O overhead during reconstruction by avoiding scattered memory writes/reads on modern hardware such as GPUs. We adopt this strategy as the standard basis selection method for BD in practice.

Let $\mathbf{I} \in \mathbb{R}^{r \times r}$ denote the identity matrix. The following identities hold for the four types of BD:

$$\text{(1) row \& first: } \mathbf{W} \equiv \begin{bmatrix} \mathbf{I} \\ \mathbf{C} \end{bmatrix} \mathbf{B}, \quad \text{(2) row \& last: } \mathbf{W} \equiv \begin{bmatrix} \mathbf{C} \\ \mathbf{I} \end{bmatrix} \mathbf{B}, \tag{2}$$
$$\text{(3) column \& first: } \mathbf{W} \equiv \mathbf{B}[\mathbf{I}, \mathbf{C}], \quad \text{(4) column \& last: } \mathbf{W} \equiv \mathbf{B}[\mathbf{C}, \mathbf{I}]$$

While the theoretical reconstruction is exact, numerical residuals may arise in practice due to finite-precision arithmetic or ill-conditioned basis matrices. To mitigate this, we compare the reconstruction errors from the first- and last-$r$ basis candidates and retain the one with the smaller Frobenius norm residual. The full procedure for *row-based* Basis Decomposition is summarized in Algorithm 4 (BD decomposition) and Algorithm 5 (BD reconstruction), where a subset of rows is selected as the basis. The *column-based* variant can be formulated analogously and is omitted here.

By losslessly replacing standard low-rank matrix multiplication with a more compact and computationally efficient alternative, BD is broadly applicable to scenarios involving low-rank multiplications. This includes applications such as neural network inference and data compression.

## 3.3 BD FOR LINEAR LAYER

The linear layer is the most common component in neural networks. Given an input vector $\mathbf{x} \in \mathbb{R}^{d_{\text{in}}}$, a standard linear layer computes the output $\mathbf{y} \in \mathbb{R}^{d_{\text{out}}}$ using a weight matrix $\overline{\mathbf{W}} \in \mathbb{R}^{d_{\text{in}} \times d_{\text{out}}}$:

$$\mathbf{y} = \mathbf{x}\overline{\mathbf{W}}. \tag{3}$$

To reduce parameter count and computational cost, many recent works adopt low-rank approximations of the weight matrix. A common approach is to factorize $\overline{\mathbf{W}}$ as $\overline{\mathbf{W}} \approx \mathbf{U}\mathbf{V}^\top$, where $\mathbf{U} \in \mathbb{R}^{d_{\text{in}} \times r}$ and $\mathbf{V} \in \mathbb{R}^{d_{\text{out}} \times r}$ with $r < \min(d_{\text{in}}, d_{\text{out}})$. The resulting low-rank layer becomes:

$$\mathbf{y} = (\mathbf{x}\mathbf{U})\mathbf{V}^\top, \tag{4}$$

which reduces both the number of parameters from $d_{\text{in}}d_{\text{out}}$ to $2r(d_{\text{in}} + d_{\text{out}})$.

Such low-rank structures appear in various domains in deep learning: (1) **low-rank pruning**, where pretrained weight matrices are compressed post hoc via SVD-like approximations (Hsu et al., 2022; Yuan et al., 2023; Wang et al., 2024; Zhao et al., 2025; Jaiswal et al., 2024; Saha et al., 2024; Kaushal et al., 2023; Sharma et al., 2023; Qinsi et al.; Liu et al., 2025; Sakr & Khailany, 2024; Ren & Zhu, 2024; Lin et al., 2024a; Hajimolahoseini et al., 2022); (2) **low-rank training**, where the weight matrices are parameterized as low-rank products throughout training (Khodak et al., 2021; Schotthöfer et al., 2022; Kamalakara et al., 2022; Zhao et al., 2023; Savostianova et al., 2023); (3) **low-rank + sparse hybridization**, which combines sparsity and low-rank approximations for improved performance (Li et al., 2023; Han et al., 2024; Zhang & Papyan, 2025); and (4) **LoRA-style fine-tuning** (Hu et al., 2022; Zhang et al., 2023b; Lialin et al., 2024; Meng et al., 2024; Liu et al., 2024c; Zhang et al., 2023a), where a low-rank adaptation is injected into frozen models for efficient parameter updates.

Since Basis Decomposition (BD) operates directly on the product $\mathbf{U}\mathbf{V}^\top$, it can be seamlessly applied to all these cases in a lossless and hardware-efficient manner.

To replace the low-rank linear layer with a BD layer, we decompose the weight product $\mathbf{W} = \mathbf{U}\mathbf{V}^\top$ using column-based BD. Let $\mathbf{B} \in \mathbb{R}^{d_{\text{in}} \times r}$ be the first-$r$ column and $\mathbf{C} \in \mathbb{R}^{r \times (d_{\text{out}} - r)}$ the coefficient matrix. The BD layer computes the output in two steps:

$$\mathbf{h} \leftarrow \mathbf{x}\mathbf{B}, \quad \mathbf{y} \leftarrow [\mathbf{h}, \mathbf{h}\mathbf{C}]. \tag{5}$$

The last-$r$ version is similar. For any $r < \min(d_{\text{in}}, d_{\text{out}})$, BD achieves strictly lower FLOPs and memory cost than the original low-rank layer, reduced by $\frac{r}{d_{\text{in}} + d_{\text{out}}}$ relative to the original.

## 3.4 BD FOR MULTI-HEAD ATTENTION

Multi-head attention (MHA) is a central component in Transformer-based architectures, widely adopted in large language models (LLMs) (Vaswani et al., 2017; Radford et al., 2018; Brown et al., 2020; Touvron et al., 2023) and vision-language models (VLMs) (Dosovitskiy et al., 2021; Radford et al., 2021; Li et al., 2022). The comparison between BD Attention and MHA is illustrated in Figure 1.

We begin by reviewing the standard multi-head attention (MHA) mechanism. Let $d$ be the input (embedding) dimension, $n$ be the number of attention heads, $d_h$ be the dimension per head, $L$ be the input sequence length, and $\mathbf{X} \in \mathbb{R}^{L \times d}$ be the attention input. MHA produces queries, keys and values ($\mathbf{Q}, \mathbf{K}, \mathbf{V} \in \mathbb{R}^{L \times nd_h}$) by three projection matrices ($\mathbf{W}_q, \mathbf{W}_k, \mathbf{W}_v \in \mathbb{R}^{d \times nd_h}$):

$$\mathbf{Q} = \mathbf{X}\mathbf{W}_q, \quad \mathbf{K} = \mathbf{X}\mathbf{W}_k, \quad \mathbf{V} = \mathbf{X}\mathbf{W}_v \tag{6}$$

$\mathbf{Q}, \mathbf{K}, \mathbf{V}$ are divided into $n$ heads for MHA:

$$[\mathbf{Q}_1, \ldots, \mathbf{Q}_n] = \mathbf{Q}, \quad [\mathbf{K}_1, \ldots, \mathbf{K}_n] = \mathbf{K}, \quad [\mathbf{V}_1, \ldots, \mathbf{V}_n] = \mathbf{V} \tag{7}$$

$$\mathbf{O}_i = \text{softmax}(\frac{\mathbf{Q}_i\mathbf{K}_i^\top}{\sqrt{d_h}})\mathbf{V}_i \tag{8}$$

$$\mathbf{Y} = [\mathbf{O}_1, \ldots, \mathbf{O}_n]\mathbf{W}_o \tag{9}$$

where $\mathbf{Q}_i, \mathbf{K}_i, \mathbf{V}_i \in \mathbb{R}^{L \times d_h}$ represent the respective query, key, and value vectors for the $i$-th head, and $\mathbf{W}_o \in \mathbb{R}^{nd_h \times d}$ represents the output projection matrix.

We reformulate multi-head attention (MHA).

$$
\begin{aligned}
\mathbf{Y} &= \sum_{i=1}^{n} \mathbf{O}_i \mathbf{W}_o^i = \sum_{i=1}^{n} \mathrm{softmax}(\frac{\mathbf{Q}_i \mathbf{K}_i^\top}{\sqrt{d_h}}) \mathbf{V}_i \mathbf{W}_o^i \\
&= \sum_{i=1}^{n} \mathrm{softmax}\left( \frac{\mathbf{X}(\mathbf{W}_q^i (\mathbf{W}_k^i)^\top) \mathbf{X}^\top}{\sqrt{d_h}} \right) \mathbf{X}(\mathbf{W}_v^i \mathbf{W}_o^i) \\
\mathbf{W}_q &\to [\mathbf{W}_q^1, \ldots, \mathbf{W}_q^n], \quad \mathbf{W}_k \to [\mathbf{W}_k^1, \ldots, \mathbf{W}_k^n], \\
\mathbf{W}_v &\to [\mathbf{W}_v^1, \ldots, \mathbf{W}_v^n], \quad \mathbf{W}_o \to \begin{bmatrix} \mathbf{W}_o^1 \\ \vdots \\ \mathbf{W}_o^n \end{bmatrix}
\end{aligned}
\tag{10}
$$

where $\mathbf{W}_q^i, \mathbf{W}_k^i, \mathbf{W}_v^i \in \mathbb{R}^{d \times d_h}$ denote the $i$-th *vertical slice* of the corresponding weight matrices, and $\mathbf{W}_o^i \in \mathbb{R}^{d_h \times d}$ denotes the $i$-th *horizontal slice* of $\mathbf{W}_o$. Both $(\mathbf{W}_q^i \mathbf{W}_k^{i\top})$ and $(\mathbf{W}_v^i \mathbf{W}_o^i)$ are a matrix with shape $d \times d_h$ multiply a matrix with shape $d_h \times d$. As $d_h < d$, this reveals a key insight:

> Each head's QK and VO computation **inherently are low-rank matrix multiplications**.

We can apply Basis Decomposition (BD) **offline during model preparation** to compress these projection weight. Taking QK as an example (VO in Appendix B), we decompose the weight product $(\mathbf{W}_q^i \mathbf{W}_k^{i\top})$ using column-based BD. Let $\mathbf{B}_{qk}^i \in \mathbb{R}^{d \times d_h}$ be the first-$r$ column (last-$r$ is similar) and $\mathbf{C}_{qk}^i \in \mathbb{R}^{d_h \times (d-d_h)}$ the coefficient matrix, we can convert the attention score into expression of BD:

$$
\begin{aligned}
\mathrm{attn\_score}_i &= \mathbf{Q}_i \mathbf{K}_i^\top = \mathbf{X}(\mathbf{W}_q^i (\mathbf{W}_k^i)^\top) \mathbf{X}^\top = \mathbf{X}(\mathbf{B}_{qk}^i [\mathbf{I}, \mathbf{C}_{qk}^i]) \mathbf{X}^\top \\
&= (\mathbf{X}\mathbf{B}_{qk}^i)([\mathbf{I}, \mathbf{C}_{qk}^i]\mathbf{X}^\top) = (\mathbf{X}\mathbf{B}_{qk}^i)(\mathbf{X}_{:,\,1:d_h}^\top + \mathbf{C}_{qk}^i \mathbf{X}_{:,\,d_h:d}^\top) \\
&= \mathbf{Q}_i' \mathbf{K}_i'^\top \\
\mathbf{Q}_i' &\leftarrow \mathbf{X}\mathbf{B}_{qk}^i \\
\mathbf{K}_i'^\top &\leftarrow \mathbf{X}_{:,\,1:d_h}^\top + \mathbf{C}_{qk}^i \mathbf{X}_{:,\,d_h:d}^\top
\end{aligned}
\tag{11}
$$

where $\mathbf{X} \to \left[ \mathbf{X}_{:,\,1:d_h}, \mathbf{X}_{:,\,d_h:d} \right]$ partitions $\mathbf{X}$ into its first $d_h$ and the remaining $d - d_h$ columns. By aligning all head's BD to either first-$r$ or last-$r$, we avoid calculating each head's Q and K projection separately, thereby reducing I/O overhead. The choice between first-$r$ and last-$r$ columns is determined by comparing the average residuals across all heads (Algorithm 3). This alignment allows Equation 11, which computes QK for a single head, to be reformulated to compute QK for all heads simultaneously:

$$
\begin{aligned}
\mathbf{Q}' &\leftarrow \mathbf{X}\mathbf{B}_{qk}, \quad \text{where } \mathbf{B}_{qk} \leftarrow [\mathbf{B}_{qk}^1, \ldots, \mathbf{B}_{qk}^n] \\
\mathbf{K}' &\leftarrow [\mathbf{X}_{:,\,1:d_h}]^{\times n} + \mathbf{X}_{:,\,d_h:d}\mathbf{C}_{qk}, \quad \text{where } \mathbf{C}_{qk} \leftarrow [(\mathbf{C}_{qk}^1)^\top, \ldots, (\mathbf{C}_{qk}^n)^\top]
\end{aligned}
\tag{12}
$$

where $[\mathbf{X}_{:,\,1:d_h}]^{\times n}$ means repeat matrix $n$ times along second dimension. VO projection could be similarly processed with BD row-based (Appendix B). We compare the original MHA inference (Algorithm 1) with MHA BD inference (Algorithm 2). The only differences (highlighted in red) are calculations of keys $\mathbf{K}$ and values $\mathbf{V}$, in which BD uses smaller size matrices for multiplication. A discussion on the impact of positional embeddings on BD is provided in Appendix D.

**Preservation of query–key similarity.** The transformed projections $\mathbf{Q}'$ and $\mathbf{K}'$ satisfy $\mathbf{Q}_i' \mathbf{K}_i'^\top = \mathbf{Q}_i \mathbf{K}_i^\top$, which means that every pairwise inner product between queries and keys is exactly preserved.

**Algorithm 1** MHA Inference

**input** Weight $\mathbf{W}_q \in \mathbb{R}^{d \times nd_h}$, $\mathbf{W}_k \in \mathbb{R}^{d \times nd_h}$, $\mathbf{W}_v \in \mathbb{R}^{d \times nd_h}$, $\mathbf{W}_o \in \mathbb{R}^{nd_h \times d}$; Input $\mathbf{X} \in \mathbb{R}^{L \times nd_h}$

1: $\mathbf{Q} \leftarrow \mathbf{X}\mathbf{W}_q$
2: $\mathbf{K} \leftarrow \mathbf{X}\mathbf{W}_k$
3: $\mathbf{V} \leftarrow \mathbf{X}\mathbf{W}_v$
4: $[\mathbf{Q}_1, \ldots, \mathbf{Q}_n] \leftarrow \mathbf{Q}$, $[\mathbf{K}_1, \ldots, \mathbf{K}_n] \leftarrow \mathbf{K}$, $[\mathbf{V}_1, \ldots, \mathbf{V}_n] \leftarrow \mathbf{V}$
5: $\mathbf{O}_i \leftarrow \text{softmax}(\frac{\mathbf{Q}_i \mathbf{K}_i^\top}{\sqrt{d_h}})\mathbf{V}_i$
6: $\mathbf{Y} \leftarrow [\mathbf{O}_1, \ldots, \mathbf{O}_n]\mathbf{W}_o$

**output** $\mathbf{Y}$

**Algorithm 2** BD Attention Inference

**input** Weight $\mathbf{B}_{qk} \in \mathbb{R}^{d \times nd_h}$, $\mathbf{C}_{qk} \in \mathbb{R}^{(d-d_h) \times nd_h}$, $\mathbf{C}_{vo} \in \mathbb{R}^{(d-d_h) \times nd_h}$, $\mathbf{B}_{vo} \in \mathbb{R}^{nd_h \times d}$; Input $\mathbf{X} \in \mathbb{R}^{L \times nd_h}$

1: $\mathbf{Q}' \leftarrow \mathbf{X}\mathbf{B}_{qk}$
2: $\mathbf{K}' \leftarrow [\mathbf{X}_{:,\,1:d_h}]^{\times n} + \mathbf{X}_{:,\,d_h:d}\mathbf{C}_{qk}$
3: $\mathbf{V}' \leftarrow [\mathbf{X}_{:,\,1:d_h}]^{\times n} + \mathbf{X}_{:,\,d_h:d}\mathbf{C}_{vo}$
4: $[\mathbf{Q}'_1, \ldots, \mathbf{Q}'_n] \leftarrow \mathbf{Q}'$, $[\mathbf{K}'_1, \ldots, \mathbf{K}'_n] \leftarrow \mathbf{K}'$, $[\mathbf{V}'_1, \ldots, \mathbf{V}'_n] \leftarrow \mathbf{V}'$
5: $\mathbf{O}'_i \leftarrow \text{softmax}(\frac{\mathbf{Q}'_i \mathbf{K}'^\top_i}{\sqrt{d_h}})\mathbf{V}'_i$
6: $\mathbf{Y} \leftarrow [\mathbf{O}'_1, \ldots, \mathbf{O}'_n]\mathbf{B}_{vo}$

**output** $\mathbf{Y}$

**Algorithm 3** BD Attention Preparation (QK)

**input** $\mathbf{W}_q, \mathbf{W}_k, \mathbf{W}_v, \mathbf{W}_o$ represent query, key, value and output projection matrix; $n$ be the number of attention heads

1: **for** $i = 1, \ldots, n$ **do**
2:   Compute first-$r$ and last-$r$ residuals for each head of QK with column-based BD:
   $R_F^i, \mathbf{B}_F^i, \mathbf{C}_F^i, R_L^i, \mathbf{B}_L^i, \mathbf{C}_L^i \leftarrow \text{BD}_{\text{col}}(\mathbf{W}_q^i \mathbf{W}_k^{i\top})$
3: **end for**
4: Compute mean residuals: $\bar{R}_F \leftarrow \frac{1}{n} \sum_{i=1}^n R_F^i$, $\bar{R}_L \leftarrow \frac{1}{n} \sum_{i=1}^n R_L^i$
5: **Select better candidate**
   **if** $\bar{R}_F \leq \bar{R}_{\text{last}}$ **then**
     $tag \leftarrow$ FIRST, $\mathbf{B}_{qk}^i \leftarrow \mathbf{B}_F^i$, $\mathbf{C}_{qk}^i \leftarrow \mathbf{C}_F^i$, $i = 1, \ldots, n$
   **else**
     $tag \leftarrow$ LAST, $\mathbf{B}_{qk}^i \leftarrow \mathbf{B}_L^i$, $\mathbf{C}_{qk}^i \leftarrow \mathbf{C}_L^i$, $i = 1, \ldots, n$

**output** $tag$, column basis matrices $\mathbf{B}_{qk}^i$ (replacement of $\mathbf{W}_q^i$), coefficient matrix $\mathbf{C}_{qk}^i$ (replacement of $\mathbf{W}_k^i$), $i = 1, \ldots, n$

$\mathbf{Q}'_i$ and $\mathbf{K}'_i$ can be regarded as an *inner-product isomorphic* representation of the original $\mathbf{Q}_i, \mathbf{K}_i$ in a $d_h$-dimensional space. We therefore still denote them as queries and keys ($\mathbf{Q}'_i, \mathbf{K}'_i$) to emphasize that they still maintain the essential property of attention: query–key similarity. Advanced compression methods relying on query–key similarity, such as KV-cache compression, remain fully compatible with BDA. Since the inner products are exactly preserved, these methods can be seamlessly integrated with BDA, enabling it to serve as a general and complementary acceleration framework.

# 4 EXPERIMENTS

We evaluate BD Attention (BDA) from three perspectives: inference accuracy and efficiency, training evaluation, and integration with low-rank pruning. All experiments are conducted on NVIDIA A6000.

## 4.1 BDA INFERENCE: ACCURACY AND EFFICIENCY

**Accuracy** Figure 2a reports the increase in perplexity on WikiText2 when replacing all MHA layers in DeepSeek-V2-Lite (16B) (Liu et al., 2024a) with BDA. Two strategies are compared: *First-r*, which always selects the first $r$ rows, and *Residual-min*, which adaptively selects between the first or last $r$ rows depending on the smaller reconstruction residual.

The perplexity increase is nearly imperceptible **0.0004% (FP32), 0.02% (FP16), 0.2% (BF16)**, with *Residual-min* consistently outperforming *First-r*. A similar advantage is observed at the operator level in per-layer reconstruction errors (see Appendix Table 4), where *Residual-min* achieves up to an order-of-magnitude lower error in FP32.

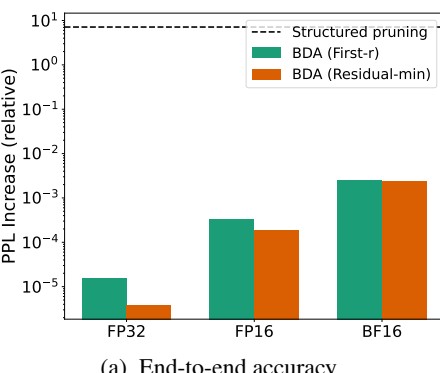

(a) End-to-end accuracy

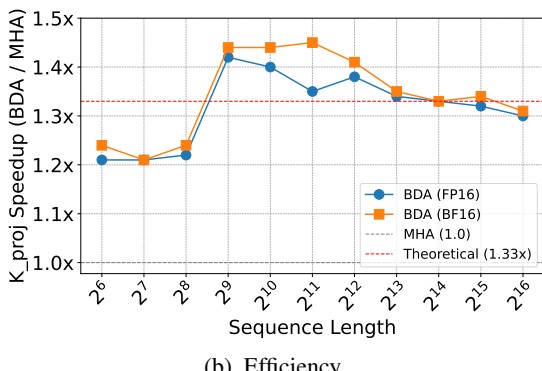

(b) Efficiency

Figure 2: **Evaluation of BD Attention (BDA). (a) End-to-end accuracy**: Perplexity ($\downarrow$) increase on WikiText2 when replacing all MHA layers of DeepSeek-V2-Lite with BDA. The increase is nearly imperceptible **0.02% (FP16)**, with *Residual-min* performing better. For reference, the dashed line shows the degradation from a structured pruning baseline at the same compression ratio (25% K/V channels). **(b) Efficiency**: Relative speedup for the $k\_proj$ operator under FP16 and BF16. The dashed line at $1.33\times$ marks the theoretical bound. Measured speedups fluctuate around this line but consistently exceed the MHA baseline, averaging **1.32×** (FP16) and **1.34×** (BF16). BDA also reduces parameter and memory usage by **25%**.

For reference, we also include a structured pruning baseline that removes 25% of K/V channels at the same compression ratio ($d_h/d = 128/512 = 25\%$). This baseline follows the relative-importance scoring strategy of Zhang et al. (2024), where each channel's importance is estimated, summed, and the least important 25% are pruned. Although more recent structured pruning techniques (Ma et al., 2023; van der Ouderaa et al., 2024; Ashkboos et al., 2024) achieve better performance, they typically require access to a calibration dataset, which is beyond the scope of this comparison. Here, structured pruning is reported only as a reference for the scale of perplexity degradation at the same compression ratio.

**Efficiency**   We adopt the DeepSeek-V3 KV configuration ($d = 512$, $d_h = 128$), where the compression ratio is $d_h/d = 25\%$. This ratio is identical across DeepSeek-V2 Lite (16B), DeepSeek-V2 (236B), and DeepSeek-V3 (660B), so the reported results generalize consistently across all these models.

To maximize efficiency, we implement a custom Triton kernel for BDA that fuses the *slice*, *repeat*, *matrix multiplication*, and *matrix addition* steps of the k_proj operator (Line 2 of Algorithm 2) into a single kernel, thereby reducing redundant memory I/O.

For comparison, we also construct a *PIFA-style attention* variant: for each head $i$, we run QR with column pivoting on $\mathbf{W}_Q^i(\mathbf{W}_K^i)^\top$ to select pivot rows (akin to the PIFA layer of Zhao et al. (2025)), then reconstruct the remaining rows via its coefficient matrix. This per-head pivoting yields different basis across heads and thus forces per-head copies and slices of $\mathbf{X}$, substantially increasing memory traffic. Consequently, *PIFA-style attention* is slower than even baseline MHA (Table 6, 7). In contrast, BDA aligns all heads to a shared contiguous basis (all first-$r$ or last-$r$), enabling a single shared $\mathbf{X}$ and coalesced memory access; with our fused Triton kernel, BDA attains near-theoretical speedups (Fig. 2b), averaging **1.32×** (FP16) and **1.34×** (BF16).

### 4.2 BDA TRAINING EVALUATION

Unlike inference, where BDA is mathematically lossless and produces outputs identical to MHA, training dynamics can differ because gradients are not guaranteed to match exactly. To evaluate this, we trained Transformer models on the IWSLT'14 English-to-German (Cettolo et al., 2014) using either standard MHA or BDA as the attention module, both under the *Noam* learning-rate schedule (Vaswani et al., 2017). We swept across four learning-rate scales ($0.5, 1, 2, 4$). As shown in Table 2, despite potential differences in optimization dynamics, the final BLEU scores of BDA are **comparable** to those of MHA across all settings. All hyperparameters are detailed in Appendix C, and

Table 2: **Training evaluation of BD Attention (BDA).** BLEU scores (↑) on the IWSLT'14 using the Transformer model. The columns correspond to the **LR scale**, i.e., the multiplicative factor applied to the learning rate of *Noam* schedule (Vaswani et al., 2017). Across all scales, BDA achieves **comparable** BLEU scores to MHA, despite not guaranteeing identical gradients, and requires no hyperparameter tuning. **Bold** numbers indicate the higher BLEU for each LR scale.

|      | LR Scale=0.5 | LR Scale=1 | LR Scale=2 | LR Scale=4 |
|------|--------------|------------|------------|------------|
| MHA  | 24.98        | 23.98      | 23.86      | 24.04      |
| BDA  | **25.27**    | **25.04**  | **23.89**  | **24.14**  |

were kept identical between MHA and BDA. This highlights that BDA requires no hyperparameter search or tuning, and thus can be *seamlessly integrated* into existing training pipelines without additional cost, while maintaining model quality.

## 4.3 BD FOR LOW-RANK PRUNING

We evaluate BD when applied on top of models already compressed by low-rank pruning (Section 3.3).

Table 3 reports results on LLaMA2-7B and LLaMA2-13B under three settings: (i) **Dense**, the original pretrained LLaMA2 model; (ii) **Low-rank (80% density)**, where weights are pruned into a low-rank structure following Zhao et al. (2025); and (iii) **BD (from low-rank)**, where the pruned low-rank weights are further transformed using Basis Decomposition (BD). No retraining is performed in any case.

Results show that BD consistently improves efficiency over the low-rank baseline while preserving perplexity. On average, BD increases throughput by **17.21%** and reduces memory usage by **16.52%** compared to low-rank models, while keeping perplexity nearly unchanged. This demonstrates that BD is complementary to existing compression techniques and can serve as a plug-in acceleration step for low-rank pruned models.

Table 3: **BD applied to low-rank pruning** on LLaMA2 models (FP16). BD further improves throughput and memory efficiency over low-rank pruning while preserving perplexity. Best throughput and memory are highlighted in **bold**.

| Model | Metric | Dense | Low rank 80% | BD (from low-rank) |
|-------|--------|-------|--------------|--------------------|
| LLaMA2-7B | Throughput (no kv cache) | 338.23 | 368.90 | **422.58** |
|  | Throughput (kv cache) | 3726.31 | 4244.27 | **5285.60** |
|  | Memory (GB) | 12.55 | 10.21 | **8.52** |
|  | PPL | 5.47 | 7.50 | 7.50 |
| LLaMA2-13B | Throughput (no kv cache) | 181.15 | 201.50 | **238.51** |
|  | Throughput (kv cache) | 2345.99 | 2566.04 | **2856.81** |
|  | Memory (GB) | 24.36 | 19.58 | **16.35** |
|  | PPL | 4.88 | 6.41 | 6.42 |

## 5 CONCLUSION AND DISCUSSION

We presented **BD Attention (BDA)**, a novel *lossless algorithmic reformulation* of multi-head attention. By applying Basis Decomposition (BD), BDA restructures projection matrices into a compact form that eliminates redundant parameters and arithmetic operations while preserving exact outputs. Our experiments confirmed its practical benefits: near-theoretical speedups in inference with reduced memory footprint, training performance comparable to MHA, and additional efficiency gains when applied to low-rank pruned models. Overall, BDA offers a mathematically exact and versatile foundation for attention acceleration that complements existing system-level methods. Future work could explore integrating BDA with FlashAttention to jointly reduce arithmetic and I/O overhead.

**Reproducibility Statement.** To ensure reproducibility, we release the source code for BDA on DeepSeek models, along with instructions for running experiment, at `https://anonymous.4open.science/r/Basis-decomp-57B8`. In addition, the full proof of Theorem 3.1 is provided in Appendix A, and the complete hyperparameter setup for BDA training experiments is detailed in Appendix C.

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

---

**Algorithm 4** BD Decomposition (Row)

---

**input** $\mathbf{W} \leftarrow \mathbf{U}\mathbf{V}^\top$, $\mathbf{W} \in \mathbb{R}^{m \times n}$ with rank $r$
 1: Extract first-$r$ rows: $\mathbf{B}_F \leftarrow$ first $r$ rows of $\mathbf{W}$
 2: Extract last-$r$ rows: $\mathbf{B}_L \leftarrow$ last $r$ rows of $\mathbf{W}$
 3: Solve coefficients:
    $\mathbf{C}_F \leftarrow \text{linsolve}(\mathbf{W}' \setminus \mathbf{B}_F = \mathbf{C}_F\mathbf{B}_F)$
    $\mathbf{C}_L \leftarrow \text{linsolve}(\mathbf{W}' \setminus \mathbf{B}_L = \mathbf{C}_L\mathbf{B}_L)$
 4: Compute residuals
$$R_F \leftarrow \left\| \mathbf{W} - \begin{bmatrix} \mathbf{B}_F \\ \mathbf{C}_F\mathbf{B}_F \end{bmatrix} \right\|_F$$
$$R_L \leftarrow \left\| \mathbf{W} - \begin{bmatrix} \mathbf{C}_L\mathbf{B}_L \\ \mathbf{B}_L \end{bmatrix} \right\|_F$$
 5: **Select better candidate**
    **if** $R_F \leq R_L$ **then**
       $tag \leftarrow$ FIRST, $\mathbf{B} \leftarrow \mathbf{B}_F$, $\mathbf{C} \leftarrow \mathbf{C}_F$
    **else**
       $tag \leftarrow$ LAST, $\mathbf{B} \leftarrow \mathbf{B}_L$, $\mathbf{C} \leftarrow \mathbf{C}_L$
**output** $tag$, basis matrix $\mathbf{B}$, coefficient matrix $\mathbf{C}$

---

---

**Algorithm 5** BD Reconstruction (Row)

---

**input** $tag$, basis matrix $\mathbf{B} \in \mathbb{R}^{r \times n}$, coefficient matrix $\mathbf{C} \in \mathbb{R}^{(m-r) \times r}$
 1: **if** $tag = $ FIRST **then** {type (1) in Equation 2}

$$\mathbf{W} \leftarrow \begin{bmatrix} \mathbf{B} \\ \mathbf{C}\,\mathbf{B} \end{bmatrix}$$

 2: **else** {type (2) in Equation 2}

$$\mathbf{W} \leftarrow \begin{bmatrix} \mathbf{C}\,\mathbf{B} \\ \mathbf{B} \end{bmatrix}$$

**output** Reconstructed matrix $\mathbf{W} \in \mathbb{R}^{m \times n}$

---

## A  PROOF OF ALMOST SURE FULL RANK OF RANDOM MATRICES THEOREM

**Theorem 3.1** (Almost Sure Full Rank of Random Matrices). *Let $\mathbf{W}$ be an $r \times r$ real random matrix. Suppose the entries of $\mathbf{W}$ are drawn from a probability measure $\mu$ on $\mathbb{R}^{r^2}$ that is absolutely continuous with respect to the Lebesgue measure $\lambda$. Then $W$ has full rank ($\text{rank}(W) = r$) with probability 1.*

*Proof.* **Step 1. Define the polynomial map $p(\mathbf{W}) = \det(\mathbf{W})$.**
Let
$$p : \mathbb{R}^{r^2} \rightarrow \mathbb{R}, \quad p(\mathbf{W}) = \det(\mathbf{W}),$$
Note that $p$ is a polynomial (in $r^2$ variables) and it is not the zero polynomial, since, for instance,
$$p(\mathbf{I}_r) = \det(\mathbf{I}_r) = 1,$$
where $\mathbf{I}_r$ denotes the $r \times r$ identity matrix.

**Step 2. The zero set of a nontrivial polynomial has measure zero.**
Consider
$$Z = \{ x \in \mathbb{R}^{r^2} \mid p(x) = 0 \}.$$
Since $p$ is a nontrivial real polynomial, its zero set $Z$ is a real algebraic variety of dimension $< r^2$, and hence we know
$$\lambda(Z) = 0.$$

(In simple terms, $Z$ is a "lower-dimensional" subset of $\mathbb{R}^{r^2}$.)

**Step 3. Absolute continuity of $\mu$ and conclusion.**
By hypothesis, the measure $\mu$ (the distribution of $\mathbf{W}$) is absolutely continuous with respect to the Lebesgue measure $\lambda$, i.e. $\mu \ll \lambda$. Hence for any Lebesgue null set $A$, we have $\mu(A) = 0$. In particular, $Z$ has $\lambda(Z) = 0$, so $\mu(Z) = 0$. But $Z$ exactly corresponds to the event $\{\det(\mathbf{W}) = 0\}$ in the space of all matrix entries. Therefore

$$\Pr(\det(\mathbf{W}) = 0) \;=\; \mu(Z) \;=\; 0,$$

which implies $\Pr(\det(\mathbf{W}) \neq 0) = 1$, i.e. $\Pr(\mathrm{rank}(\mathbf{W}) = r) = 1$. $\qquad\square$

## B  BD FOR VO IN MHA

Here, we introduce the BD transformation for value and output projection matrices. The right part of Equation 10 can be converted to (using row & first in Equation 2):

$$\mathbf{V}_i\mathbf{W}_o^i = \mathbf{X}(\mathbf{W}_v^i\mathbf{W}_o^i) = \mathbf{X}(\begin{bmatrix}\mathbf{I}\\\mathbf{C}_{vo}^i\end{bmatrix}\mathbf{B}_{vo}^i) = (\mathbf{X}_{:,\,1:d_h} + \mathbf{X}_{:,\,d_h:d}\mathbf{C}_{vo}^i)\mathbf{B}_{vo}^i \tag{13}$$

where $\mathbf{B}_{vo}^i \in \mathbb{R}^{d_h \times d}$ be the first-$d_h$ row basis and $\mathbf{C}_{vo}^i \in \mathbb{R}^{(d-d_h)\times d_h}$ the coefficient matrix. Similar to Equation 12, we can redefine value and output projection matrices to avoid calculating each head separately:

$$
\begin{aligned}
\mathbf{V}' &= [\mathbf{X}_{:,\,1:d_h}]^{\times n} + \mathbf{X}_{:,\,d_h:d}\mathbf{C}_{vo}, \quad \text{where } \mathbf{C}_{vo} := [\mathbf{C}_{vo}^1, \ldots, \mathbf{C}_{vo}^n]\\
[\mathbf{V}'_1, \ldots, \mathbf{V}'_n] &= \mathbf{V}'\\
\mathbf{O}'_i &= \mathrm{softmax}(\frac{\mathbf{Q}'_i{\mathbf{K}'_i}^\top}{\sqrt{d_h}})\mathbf{V}'_i\\
\mathbf{Y} &= [\mathbf{O}'_1, \ldots, \mathbf{O}'_n]\mathbf{B}_{vo}, \quad \text{where } \mathbf{B}_{vo} := \begin{bmatrix}\mathbf{B}_{vo}^1\\\vdots\\\mathbf{B}_{vo}^n\end{bmatrix}
\end{aligned}
\tag{14}
$$

where $[\mathbf{X}_{:,\,1:d_h}]^{\times n}$ means repeat matrix $n$ times along second dimension. The *last* version is similar.

## C  HYPERPARAMETERS FOR BDA TRAINING

For the IWSLT'14 English-to-German task, we followed the standard Transformer setup with the *Noam* learning-rate schedule (Vaswani et al., 2017). The embedding dimension was set to 512, with a feed-forward dimension of 2048. We used a batch size of 10,240 tokens and trained for 20,000 steps. Dropout was 0.1 and label smoothing was 0.1. The model contained 6 layers with 4 attention heads. For simplicity, we removed positional embeddings inside the MHA module but retained them in the embedding layer (the effect of positional embeddings on BD is discussed in Appendix D). The learning rate scale was varied in $\{0.5, 1, 2, 4\}$ (Section 4.2). Training employed the Adam optimizer with 6,000 warmup steps. Beam search with a beam size of 2 was used for evaluation, and BLEU scores were reported using the checkpoint with the lowest validation perplexity.

All hyperparameters for BDA were kept **identical** to those of MHA to ensure a fair comparison. Moreover, this also shows that BDA naturally matches the training dynamics of MHA, achieving comparable performance without any hyperparameter tuning. As a result, existing LLM training pipelines can migrate from MHA to BDA at essentially no cost.

## D  EFFECT OF POSITIONAL EMBEDDING ON BD

We briefly discuss how positional embeddings interact with Basis Decomposition (BD).

**Embedding-layer positional embedding.** Any positional embedding applied at the embedding layer (e.g., learned positional embeddings or sinusoidal embeddings added to input tokens) does not affect BD. Since BD only restructures the projection matrices inside the attention, the addition of position-dependent vectors at the input embedding layer is orthogonal to BD's reformulation.

**MHA-internal positional embedding.** When positional embeddings are applied inside the multi-head attention (MHA) module, the formulation changes. For example, vanilla Rotary Position Embedding (RoPE) (Su et al., 2021) modifies the attention score computation as

$$\mathbf{X}_n \mathbf{W}_q \, \mathbf{W}_k^\top \mathbf{X}_m^\top \quad \longrightarrow \quad \mathbf{X}_n \mathbf{W}_q \, \mathbf{R}_{n-m} \, \mathbf{W}_k^\top \mathbf{X}_m^\top$$

where $\mathbf{R}_{n-m}$ is a rotation matrix depending on relative positions $n - m$. BD guarantees the exact factorization

$$\mathbf{W}_q \mathbf{W}_k^\top = \mathbf{B}[\mathbf{I}, \mathbf{C}],$$

but in general BD cannot guarantee

$$\mathbf{W}_q \mathbf{R}_{n-m} \mathbf{W}_k^\top = \mathbf{B} \mathbf{R}_{n-m} [\mathbf{I}, \mathbf{C}].$$

Thus, vanilla RoPE breaks the exactness of BD.

**Decoupled RoPE as a solution.** DeepSeek proposes *Decoupled RoPE* (Liu et al., 2024a), which splits attention channels into RoPE and non-RoPE parts. BD can then be applied to the non-RoPE channels, while RoPE channels remain unchanged. This is also the strategy used in our experiments.

**Model-specific implications.**

- **GPT models** use positional embeddings only at the input embedding layer; thus BD is fully lossless for both QK and VO.
- **LLaMA models** adopt vanilla RoPE inside MHA. Since RoPE is applied only to QK, BD remains lossless for VO projections but is not exact for QK.
- **DeepSeek models** employ Decoupled RoPE, which separates RoPE and non-RoPE channels in QK. BD can be applied losslessly to the non-RoPE channels of QK, and to all VO projections.

In summary, the compatibility of BD with positional embeddings depends on how positional information is integrated. Embedding-layer positional encodings pose no issue, while RoPE inside MHA requires modifications (e.g., Decoupled RoPE).

# E   LLM USAGE

Large Language Models (LLMs) were used solely as a writing assistant to polish grammar. They were not involved in research ideation, experiment design, analysis, or drafting of scientific content.

Table 4: **Numerical reconstruction errors of BD** for $\mathbf{W}_q\mathbf{W}_k^\top$ (QK) and $\mathbf{W}_v\mathbf{W}_o^\top$ (VO) under different floating-point formats. Values are averaged across all heads and all layers of the DeepSeek-V2-Lite model. We compare two strategies for selecting the BD basis: (i) always using the first $r$ rows (*First-r*), and (ii) choosing between the first or last $r$ rows depending on which yields the smaller residual (*Residual-min*). We report absolute mean squared error (MSE) and normalized mean squared error (NMSE). The results confirm that BD introduces only **negligible** perturbations to the matrix products, with *Residual-min* consistently outperforming *First-r*, and improving error by at least one order of magnitude in FP32.

|  |  | FP32 | FP16 | BF16 |
|---|---|---|---|---|
| QK MSE | First-$r$ | $3.19 \times 10^{-12}$ | $1.09 \times 10^{-7}$ | $7.42 \times 10^{-7}$ |
|  | Residual-min | $3.12 \times 10^{-13}$ | $7.51 \times 10^{-8}$ | $6.51 \times 10^{-7}$ |
| QK NMSE | First-$r$ | $5.74 \times 10^{-9}$ | $3.20 \times 10^{-4}$ | $2.07 \times 10^{-3}$ |
|  | Residual-min | $7.10 \times 10^{-10}$ | $2.36 \times 10^{-4}$ | $1.88 \times 10^{-3}$ |
| VO MSE | First-$r$ | $2.45 \times 10^{-12}$ | $1.02 \times 10^{-8}$ | $8.91 \times 10^{-8}$ |
|  | Residual-min | $2.15 \times 10^{-14}$ | $5.97 \times 10^{-9}$ | $8.19 \times 10^{-8}$ |
| VO NMSE | First-$r$ | $1.26 \times 10^{-7}$ | $2.71 \times 10^{-4}$ | $2.28 \times 10^{-3}$ |
|  | Residual-min | $8.31 \times 10^{-10}$ | $1.61 \times 10^{-4}$ | $2.06 \times 10^{-3}$ |

Table 5: **End-to-end evaluation of BD Attention on WikiText2**. All MHA layers in the DeepSeek-V2-Lite model are replaced with BD Attention, and we report perplexity ($\downarrow$) under different floating-point formats. For BD, we show results using (i) always the first $r$ rows (*First-r*) and (ii) selecting between the first or last $r$ rows based on the smaller residual (*Residual-min*). The last row reports the *relative increase in perplexity (PPL)*, computed as $(\text{PPL}_{\text{BD}} - \text{PPL}_{\text{Original}})/\text{PPL}_{\text{Original}}$. Across all settings, BD introduces only **negligible** changes in model performance, with *Residual-min* consistently yielding smaller increases. Preparation time is very short (a few seconds), making BD efficient for deployment.

|  |  | FP32 | FP16 | BF16 |
|---|---|---|---|---|
| Original PPL |  | 6.306983 | 6.307075 | 6.310289 |
| BD PPL | First-$r$ | 6.307082 | 6.309186 | 6.326459 |
|  | Residual-min | 6.307007 | 6.308252 | 6.325656 |
| PPL Increase (relative) | First-$r$ | 0.002% | 0.033% | 0.256% |
|  | Residual-min | 0.0004% | 0.019% | 0.244% |
| Preparation Time (s) | First-$r$ | 3.56 | 1.89 | 2.39 |
|  | Residual-min | 6.09 | 4.05 | 4.05 |

Table 6: **Throughput comparison on FP16, NVIDIA A6000**. We report throughput in million tokens per second (higher is better) for a single attention operator, comparing MHA, PIFA-style Attention, and BDA across different sequence lengths. The tested matrix shape follows the DeepSeek-V3 configuration (Liu et al., 2024b), with $n = 128$ heads, $d = 512$ (corresponding to $d_c$ in DeepSeek-V3), and $d_h = 128$ (same as $d_h$ in DeepSeek-V3). The last column reports the *relative speedup*, defined as BDA throughput divided by MHA throughput.

| Seq. Len | MHA | PIFA-style (per-head QR) | BDA | Speedup |
|---|---|---|---|---|
| 64 | 1.79 | 0.99 | 2.16 | 1.21× |
| 128 | 3.13 | 1.30 | 3.79 | 1.21× |
| 256 | 4.46 | 1.52 | 5.43 | 1.22× |
| 512 | 4.95 | 1.51 | 7.04 | 1.42× |
| 1024 | 5.62 | 1.69 | 7.87 | 1.40× |
| 2048 | 5.95 | 1.72 | 8.03 | 1.35× |
| 4096 | 5.59 | 1.72 | 7.71 | 1.38× |
| 8192 | 5.58 | 1.74 | 7.47 | 1.34× |
| 16384 | 5.51 | 1.74 | 7.31 | 1.33× |
| 32768 | 5.43 | 1.74 | 7.17 | 1.32× |
| 65536 | 5.41 | 1.72 | 7.06 | 1.30× |

Table 7: **Throughput comparison on BF16, NVIDIA A6000**. Setup and notation are the same as Table 6.

| Seq. Len | MHA | PIFA-style (per-head QR) | BDA | Speedup |
|---|---|---|---|---|
| 64 | 1.74 | 0.98 | 2.16 | 1.24× |
| 128 | 3.13 | 1.26 | 3.79 | 1.21× |
| 256 | 4.46 | 1.52 | 5.56 | 1.24× |
| 512 | 4.95 | 1.50 | 7.14 | 1.44× |
| 1024 | 5.62 | 1.72 | 8.06 | 1.44× |
| 2048 | 5.62 | 1.72 | 8.13 | 1.45× |
| 4096 | 5.59 | 1.73 | 7.89 | 1.41× |
| 8192 | 5.61 | 1.74 | 7.60 | 1.35× |
| 16384 | 5.52 | 1.74 | 7.34 | 1.33× |
| 32768 | 5.50 | 1.74 | 7.34 | 1.34× |
| 65536 | 5.51 | 1.72 | 7.22 | 1.31× |

