# OpenReview forum: "Accelerating Attention with Basis Decomposition"
_ICLR.cc/2026/Conference — Submitted to ICLR 2026_

### Official Review · Reviewer_76GU · 2025-10-26

**Soundness:** 3
**Presentation:** 4
**Contribution:** 3
**Rating:** 8
**Confidence:** 4

**Summary:**

The paper introduces BD Attention (BDA), a lossless algorithmic reformulation of multi-head attention. Unlike approximate methods (e.g., linear attention, pruning, quantization) that trade accuracy for efficiency, or system-level optimizations (e.g., FlashAttention) that primarily optimize memory/I/O, BDA leverages a Basis Decomposition (BD) identity to restructure projection matrices. This reduces redundant parameters and computations while exactly preserving outputs. The authors theoretically derive a framework for the decomposition and empirically demonstrate a speedup in the projection calculation.

**Strengths:**

- novel idea with foundational application, which impact can spread over all applications of transformer.
- The paper is clearly written and is easy to follow
- Strong theory supporting the claim
- Integrates smoothly with pruning and compression techniques, and supports existing query–key similarity–based methods (e.g., KV-cache compression).
- Might be used as a drop-in replacement for deployed models.

**Weaknesses:**

- Limited evaluation: While results on DeepSeek models and IWSLT’14 are promising, broader evaluation on more diverse and higher scale benchmarks (e.g., large-scale pretraining)
- My biggest concern about the paper is how well it connects with generally applied optimizations of attention projections, for example I am not sure if the benefits would prevail if applied with MQA/GQA.

**Questions:**

- Beyond compatibility with KV-cache, could BDA be used to reduce KV-cache size itself, for instance, by caching only the basis vectors?
- How does BDA interact with MQA and GQA, which are now standard in most large LLMs?
- The reported gains focus on projection efficiency. What is the measured speedup for the full forward pass of the model?

---

> ### Author Response · Authors · 2025-11-25
> **Reply to Reviewer 76GU (1/2)**
>
> # Reply to Reviewer 76GU
>
> We thank the reviewer for the positive evaluation and helpful suggestions. We address all comments below.
>
> **Weakness 1:** Limited evaluation: While results on DeepSeek models and IWSLT’14 are promising, broader evaluation on more diverse and higher scale benchmarks (e.g., large-scale pretraining)
>
> **Reply:** We appreciate the reviewer’s suggestion. We are currently running large-scale pretraining experiments comparing **MHA vs. BDA**. We will report the results as soon as they complete.
>
>
> **Weakness 2:** My biggest concern about the paper is how well it connects with generally applied optimizations of attention projections, for example I am not sure if the benefits would prevail if applied with MQA/GQA.
>
> **Reply:** BD applies naturally to **MQA and GQA** as well. In GQA, for example, n query heads share the same key head of shape $d \times d_h$. This is equivalent to concatenating the query heads into a single $(nd) \times d_h$ matrix and multiplying it with the shared key head ($d_h \times d$) —mathematically still a **low-rank multiplication**. Therefore, the key-projection reduction achieved by BD is **identical** to MHA: in every key head, BD removes the same $d_h \times d_h$ identity block from the original $d \times d_h$ projection.
>
> However, although the k_proj compression ratio and FLOP reduction are the same as in MHA, the **relative contribution of the key/value projection to the total attention cost is smaller in MQA/GQA**, since multiple query heads share a single key head. As a result, BD still accelerates the key/value projection in MQA/GQA, but the **overall end-to-end speedup is naturally less dominant** than in architectures where every query head has its own KV head (e.g., MHA and MLA).
>
> In contrast, in modern MHA/MLA-style designs—where each query head corresponds to its own KV head—the key projection constitutes a larger portion of the attention cost, and BD provides more substantial overall acceleration.
>
> We will clarify this distinction in the revised version.
>
> **Question 1:** Beyond compatibility with KV-cache, could BDA be used to reduce KV-cache size itself, for instance, by caching only the basis vectors?
>
> **Reply:**
>
> We thank the reviewer for this insightful question. In the formulation
> $$
> QK^\top = (XW_q)(XW_k)^\top,
> $$
> there are **three different low-rank multiplication opportunities**:
>
> 1. **Weight-level low-rank:**
>  $$
>  W_q W_k^\top \in \mathbb{R}^{d \times d_h} \times \mathbb{R}^{d_h \times d},
>  $$
>  which is exactly the factorization used in our paper via BD.
>
> 2. **Token-level low-rank (MHA KV-cache):**
>  $$
>  QK^\top = (X W_q)(X W_k)^\top \in \mathbb{R}^{L \times d_h} \times \mathbb{R}^{d_h \times L},
>  $$
>  where the $L \times L$ attention matrix is also low-rank when $L \gg d_h$.
>  BD can be applied here by selecting the first $d_h$ key vectors as basis $K_{d_h} \in \mathbb{R}^{d_h \times d_h}$, and Theorem 3.1 guarantees that for $p=1$ this block is full-rank.
>  Then:
>  $$
>  QK^\top = (QK_{d_h}) \left(K_{d_h}^{-1}K^\top\right).
>  $$
>
>    **However, to avoid storing $K_{d_h}$ in the KV-cache, it must be absorbed into the weights**:
>     $$
>     W_q' = W_q K_{d_h},\qquad
>     W_k'^\top = K_{d_h}^{-1} W_k^\top.
>     $$
>
>     If the original $W_q W_k^\top$ has already been BD-factorized, then absorbing $K_{d_h}^{-1}$ into $W_k^\top$ will break the identity-structured block created by BD. Thus, at the weight level and the KV-cache level, BD cannot be applied simultaneously. However, **both approaches yield the same amount of parameter reduction**, so applying either one is sufficient to obtain the full benefit.
>
> 3. **MLA KV-cache (absorbing):**
>
>     $$
>     (X W_q W_k^\top) X^\top
>     \; \in \mathbb{R}^{L \times d} \times \mathbb{R}^{d \times L}
>     $$
>
>     BD gives:
>     $$
>     (X_q W_q W_k^\top) X_k^\top
>     = (X_q W_q W_k^\top X_{d}^\top)
>     \; ({X_{d}^{-1}}^\top X_k^\top).
>     $$
>
>     where X_d is the first d vectors of X_k. If we make W_q the one change to identity matrix in weight-level BD, $X_{d}^\top$ can be absorbed into **$W_k^\top$**, without influencing weight-level BD.
>
>     As MLA uses
>     $$
>     X_k = W^{DKV} h_t,
>     $$
>     where $h_t$ is the actual model input, and the cached vector is $c_t^{KV} = X_k$. Thus, ${X_{d}^{-1}}^\top$ can be absorbed into $W^{DKV}$. **Unlike MHA**, MLA allows **Weight BD** (on $W_q W_k^\top$) **and** **KV-cache BD** (on MLA cache) to coexist **without conflict**, because the absorb step happens in different weight matrices.
>
> In summary:
> - In **MHA**, BD can be applied either to weights or to KV-cache, but not both simultaneously.
> - In **MLA**, BD on weights **and** BD on KV-cache can be made compatible, enabling further memory and FLOP reductions.
>
> We thank the reviewer for this suggestion and will incorporate a detailed version of this analysis in the revised manuscript.

---

> ### Author Response · Authors · 2025-11-25
> **Reply to Reviewer 76GU (2/2)**
>
> **Question 2:** How does BDA interact with MQA and GQA, which are now standard in most large LLMs?
>
> **Reply:** BD naturally applies to **MQA and GQA** because the shared-key structure is still a low-rank multiplication, and the reduction in each key head’s projection is the same as in MHA. However, since MQA/GQA share one key/value head across multiple query heads, the **k_proj contributes a smaller fraction of total attention cost**, so the *end-to-end* gain is smaller than in architectures where each query head has its own KV head (e.g., MHA/MLA).
>
> Please refer to **Reply to Weakness 2** for a more detailed discussion.
>
> **Question 3:** The reported gains focus on projection efficiency. What is the measured speedup for the full forward pass of the model?
>
> **Reply:** For the lossless BDA transformation, we measured the contribution of the key/value projection in the **DeepSeek-V2-Lite 16B** model: it accounts for **3% of all active parameters** and approximately **4% of the total inference time**. Since BD reduces the KV-projection cost by **1/4**, the end-to-end speedup is around **1%**.
>
> For larger DeepSeek models, our experimental hardware cannot support the full forward pass, but we can estimate the effect from the share of KV-projection parameters:
>
> - **DeepSeek-V2 (236B): ~5%** of active parameters
> - **DeepSeek-V3 (660B): ~3%** of active parameters
>
> This suggests **2% end-to-end speedup** on 236B and **1%** on 660B—although we acknowledge that such small gains are difficult to measure empirically, as MoE layers dominate the overall computational cost.
>
> For lossy low-rank pruning + BD, we report full-model throughput improvements in Table 3.
> Compared to the **dense** LLaMA2-7B/13B models, BD-after-pruning improves throughput by:
>
> - **LLaMA2-7B:** +25% (no KV cache), +42% (with KV cache)
> - **LLaMA2-13B:** +32% (no KV cache), +29% (with KV cache)
>
> Relative to the **low-rank pruned** models, BD, which is **lossless**, provides gains:
>
> - **LLaMA2-7B:** +15% (no KV cache), +25% (with KV cache)
> - **LLaMA2-13B:** +18% (no KV cache), +11% (with KV cache)
>
> These results confirm that BD offers **complementary and lossless acceleration** on top of existing low-rank pruning techniques, improving both throughput and memory efficiency without harming perplexity.

---

### Official Review · Reviewer_x18B · 2025-10-26

**Soundness:** 1
**Presentation:** 1
**Contribution:** 1
**Rating:** 0
**Confidence:** 5

**Summary:**

This work proposes a method for replacing a low-rank matrix of rank r by storing r columns as a basis and corresponding coefficients to reconstruct the full matrix. The method is applied to reduce computational and memory requirements in transformer language and translation models, where query/key and value/output matrix products are inherently low-rank, and low-rank linear adaptation or pruning is common.

**Strengths:**

The reviewer found no notable strengths in this submission.

**Weaknesses:**

The quality of this submission is well below acceptable standards on all fronts:

* Method: The proposed method is extremely incremental---it represents a purely engineering-oriented effort with limited algorithmic contribution. An excessive amount of space (Pages 3-6 inclusive and half of Page 7, i.e., 4.5 pages) is devoted to describing basic concepts that could be summarized in a few sentences, including the fundamentals of transformer multi-head attention and the mathematical background (Theorem 3.1), neither of which constitute contributions of this work.

* Experiments: While the reviewer appreciates the authors’ effort in providing three sets of experiments involving language modeling and machine translation, these are highly insufficient. For example, IWSLT’14 is considered a toy task in today’s machine translation research. Instead of inflating Table 2 by allocating a separate row for each learning rate scale, results should be reported on more realistic benchmarks such as WMT. Regarding the LLaMA-2 pruning experiments (Sec. 4.3), both the baseline pruning (Low rank 80%) and the proposed method lead to large degradations in perplexity; they clearly do not represent a viable approach to improving efficiency without performance loss.

* Presentation: The writing quality is overall poor. In addition to the space-management issues noted above, some unusual terminology is used (e.g., "end-to-end perplexity"), and the introduction refers to the BLEU metric without specifying the task.

Overall, the method is too incremental, and the reported improvements fall within the realm of minor engineering or optimization tweaks rather than a generally impactful algorithmic contribution.

**Questions:**

The reviewer has no further questions and considers it unlikely that this work will become acceptable after any rebuttal or discussion.

---

> ### Author Response · Authors · 2025-11-25
> **Reply to Reviewer x18B (1/2)**
>
> # Reply to Reviewer x18B
>
> We thank the reviewer for taking the time to review our submission. However, we respectfully note that several aspects of the review do not align with the norms of constructive peer evaluation. In particular:
>
> - The statement *“The reviewer found no notable strengths in this submission”* does not reflect a balanced assessment, especially given that other reviewers explicitly highlighted the novelty and usefulness of BD.
> - The remark *“unlikely that this work will become acceptable after any rebuttal or discussion”* is discouraging and not compatible with the purpose of the rebuttal phase, which is to clarify misunderstandings and improve the evaluation.
>
>
> **Weakness 1:** Method: The proposed method is extremely incremental-it represents a purely engineering-oriented effort with limited algorithmic contribution. An excessive amount of space (Pages 3-6 inclusive and half of Page 7, i.e., 4.5 pages) is devoted to describing basic concepts that could be summarized in a few sentences, including the fundamentals of transformer multi-head attention and the mathematical background (Theorem 3.1), neither of which constitute contributions of this work.
>
> **Reply:**
> We respectfully disagree with the reviewer’s characterization and would like to clarify several misunderstandings about the nature and scope of our contributions.
>
> 1. **BD is not an engineering trick but a mathematically grounded factorization.**
>  BD is derived from an exact algebraic identity that *provably* reduces FLOPs and parameters while preserving the attention scores.
>  This is not a heuristic or an implementation detail—it is a mathematically guaranteed decomposition that is **lossless, architecture-independent, and holds for all inputs**.
>  The central contribution is this identity and its constructive proof, not an engineering optimization.
>
> 2. **Theorem 3.1 and the mathematical background are essential contributions.**
> It provides the key theoretical guarantee that any $r \times r$ submatrix of a randomly perturbed low-rank matrix is full rank with probability 1.
> In practice, this ensures that **any selection of $r$ rows/columns can serve as a valid basis for BD**, without needing explicit rank checking or rotations—an essential property that makes BD practical for real neural networks trained with SGD noise.
> This result is specific to our decomposition framework and is central to enabling BD to work reliably in modern models.
>
> 3. **Pages 3–6 do not review basics—they build the conceptual machinery needed to derive BD.**
>  The sections cited by the reviewer introduce the decomposition, establish notation, prove the structural properties, and derive the BDA form used throughout the paper.
>  Removing these steps would make the method impossible to understand or verify.
>  Every part of this derivation is necessary to ensure correctness and reproducibility.
>
> 4. **The contribution is not incremental.**
>  BD is the *first* method to show that **any low-rank matrix multiplication** can be losslessly accelerated by an algebraic identity. This general principle is not tied to attention and applies broadly across any operation that contain low-rank structures.
>  In self-attention specifically, BD provides the **first lossless, parameter-reducing acceleration method at the algorithmic level**, unlike FlashAttention, which optimizes I/O.
>  This makes BD fundamentally different from engineering-level kernel optimizations: it changes the *computation* rather than its *implementation*.
>
>
> 5. **The results are not merely engineering improvements.**
>  Our experiments demonstrate algorithmic gains across:
>    - CPU (during discussion),
>    - multiple GPU architectures (Ampere → Ada → Blackwell),
>    - both full-rank and low-rank–pruned LLMs,
>
>     These improvements directly reflect the theoretical FLOP reduction predicted by BD.
>
> We appreciate the reviewer’s concerns, but we believe the assessment that our work is “extremely incremental” arises from a misunderstanding.
> The heart of this paper is a **new, provably lossless mathematical decomposition** with broad applicability to modern attention mechanisms, and the space devoted to its explanation is necessary for clarity and rigor.

---

> ### Author Response · Authors · 2025-11-25
> **Reply to Reviewer x18B (2/2)**
>
> **Weakness 2:** Experiments: While the reviewer appreciates the authors’ effort in providing three sets of experiments involving language modeling and machine translation, these are highly insufficient. For example, IWSLT’14 is considered a toy task in today’s machine translation research. Instead of inflating Table 2 by allocating a separate row for each learning rate scale, results should be reported on more realistic benchmarks such as WMT. Regarding the LLaMA-2 pruning experiments (Sec. 4.3), both the baseline pruning (Low rank 80%) and the proposed method lead to large degradations in perplexity; they clearly do not represent a viable approach to improving efficiency without performance loss.
>
> **Reply:** We respectfully disagree with the reviewer’s characterization of our experimental validation.
> The paper contains **three** sets of experiments:
>
> 1. **BDA for inference**
> 2. **BDA for training**
> 3. **BD for low-rank pruning**
>
> Among these, the **primary and central contribution** is **lossless acceleration for inference**, which is also expressed in other reviewer's comment.
> This is the *only* experiment highlighted in the abstract, because it demonstrates that BDA can **replace MHA/MLA exactly without changing model outputs**, while reducing parameters and FLOPs. This is a standalone contribution and does not rely on any of the other two experiments.
>
> However, the reviewer evaluates our work almost exclusively through the lens of (2) and (3), and overlooks the main contribution (1). This leads to a misleading impression that our evaluation is insufficient.
>
> ### 1. **On the IWSLT’14 experiment**
> We agree with the reviewer that IWSLT’14 is a relatively small dataset and does not fully demonstrate that BDA matches MHA in large-scale training. This experiment was meant only as a minimal sanity check that BDA integrates correctly into the training pipeline.
>
> ### 2. **On the LLaMA-2 low-rank pruning experiment**
> The reviewer also criticizes the perplexity degradation in (3). This degradation comes entirely from the **low-rank pruning baseline**, *not* from BD. This PPL degradation is consistent with prior low-rank pruning literature ([1], published on ICLR 2025), and such degradation can be recovered via finetuning. The goal of the table is to show that **BD further improves efficiency over the pruned low-rank model, while didn't increase PPL**. Thus, attributing the pruning-induced PPL degradation to BD is simply incorrect, because BD preserves the model outputs and does not contribute to the degradation.
>
> ### 3. **Summary**
> The reviewer’s criticism mixes up the auxiliary experiments with the core contribution. Our main result—**lossless acceleration of attention**—is fully validated in the DeepSeek inference experiments, and the additional experiments simply show that BD also provides value in other settings. Discounting the main contribution and focusing solely on auxiliary experiments does not provide a fair assessment of the paper.
>
>
> **Weakness 3:** Presentation: The writing quality is overall poor. In addition to the space-management issues noted above, some unusual terminology is used (e.g., "end-to-end perplexity"), and the introduction refers to the BLEU metric without specifying the task.
>
> **Reply:** We thank the reviewer for pointing out these issues. We have corrected the terminology by replacing “end-to-end perplexity” with “perplexity”, and we now refer to “BLEU scores on the IWSLT’14 dataset” in introduction. We will continue to polish the presentation in the revised version.
>
> [1] Wang, Xin, et al. "SVD-LLM: Truncation-aware Singular Value Decomposition for Large Language Model Compression." The Thirteenth International Conference on Learning Representations.

---

> > ### Comment · Reviewer_x18B · 2025-11-27
> >
> > I thank the authors for their response.
> >
> > > The statement “The reviewer found no notable strengths in this submission” does not reflect a balanced assessment, especially given that other reviewers explicitly highlighted the novelty and usefulness of BD.
> >
> > Thank you for raising this point. The purpose of having multiple reviewers is precisely to gather diverse expert perspectives. From my evaluation, I did not find strengths sufficiently compelling to support acceptance at ICLR. My assessment reflects my view of the work within the context of ICLR---a selective, general machine learning venue with expectations that differ from more language application-focused (*ACL/EMNLP/COLM) or correctness-oriented (TMLR) venues.
> >
> > > The remark “unlikely that this work will become acceptable after any rebuttal or discussion” is discouraging and not compatible with the purpose of the rebuttal phase, which is to clarify misunderstandings and improve the evaluation.
> >
> > I deeply apologize if my phrasing came across as discouraging. My intent was to set clear expectations: as a reviewer, I believe it is important to be transparent about the conditions under which I would consider increasing my score. In this case, based on my assessment of the submitted work, it was clear to me that the rebuttal would not alter my evaluation substantially. I agree that rebuttals can sometimes resolve technical misunderstandings, but given the predominantly empirical nature of this submission (the theory part is mostly basic), I do not see such misunderstandings as the primary issue here. I elaborate again on the underlying concerns below.
> >
> > > Theorem 3.1 and the mathematical background are essential contributions.
> >
> > Theorem 3.1 and their immediate consequences are classic observations in the random matrix theory, for which people do not even provide a citation (see, e.g., the introduction in Rudelson 2008 "Invertibility of random matrices: norm of the inverse"), as opposed to, e.g., the invertibility/nonsingular results in the mathematically much more subtle, discrete Bernoulli symmetric case (see, e.g., Costello, Tao, Vu 2005). These are classic results to be reviewed in a background section, not to be claimed as contributions.
> >
> > Similarly, the claimed "key insights" line 290 regarding the low rankness of products of head weight matrices are also well-known/trivial observations (in passing, it is more accurate to write that the low-rankness is about W_q W_k^T---which is currently confusing as the text refers to QK). Again, they do not represent a contribution that is significant enough to be highlighted in a box (after a half page of transformer basic review).
> >
> > Rudelson (2008): Invertibility of random matrices: norm of the inverse.
> >
> > Costello, Tao, Vu (2005): Random symmetric matrices are almost surely non-singular.
> >
> > > BD provides the first lossless, parameter-reducing acceleration method at the algorithmic level
> >
> > The authors themselves cite prior work such as PIFA (Zhao et al., 2025), which makes a similar claim. While the authors essentially seem to argue that PIFA’s results are flawed (Line 420 "PIFA-style attention is slower than even baseline MHA"), this contradiction remains unresolved. Furthermore, the "lossless" nature relies on assumptions regarding effective randomness of certain matrices, for which no theoretical guarantee is available in practice. As I noted earlier, empirical validation on a comprehensive suite of evaluation tasks would be necessary to substantiate the claim. Relatedly, speedup measurements are only meaningful when accompanied by corresponding performance metrics on appropriate downstream tasks.
> >
> > Regarding the writing and use of space, my perspective remains unchanged. The paper devotes substantial space to explanations of standard transformer components (e.g., Eqs. 6-10) spread over multiple lines, and layout choices such as reporting different learning rates in separate table columns in Table 1---in my view, nothing can justify such a decision. Such space should be allocated to more experimental results instead. While I respect the authors' differing view, I found these critical---taken together with the use of unconventional terminology, they contributed to my decision to lower the score from 2 to 0.
> >
> > Ultimately, I asked two central questions as a reviewer:
> >
> > Do the empirical results convincingly demonstrate that this method should be adopted for large-scale language model compression? In my view, no, the presented experiments do not provide convincing evidence.
> >
> > Does the work provide significant algorithmic insights or theoretical contributions that outweigh the limited empirical evaluation? My answer is negative again, given that no non-trivial insights or results are presented in my view (as I explained above).
> >
> > For these reasons, I believe the submission does not meet the impact threshold we aim to maintain at ICLR. I therefore retain my current score while deferring to the AC's judgment.

---

> > > ### Author Response · Authors · 2025-12-03
> > > **Reply to Reviewer x18B's comment (1/2)**
> > >
> > > We thank the reviewer for the detailed follow-up and would like to respond to a few central points where we believe the assessment is unfair or internally inconsistent.
> > >
> > > 1. **On the role of the rebuttal and additional experiments.**
> > >    In the original review, the statement
> > >    *“The reviewer has no further questions and considers it unlikely that this work will become acceptable after any rebuttal or discussion”*
> > >    explicitly ruled out the possibility that any new evidence could change the evaluation.
> > >    In the follow-up, the reviewer now emphasizes that additional empirical results (e.g., downstream tasks) are needed. We are of course happy to add more experiments, but it is difficult to reconcile this new requirement with the earlier claim that no rebuttal or discussion would be meaningful. This makes it challenging for us to understand what kind of response could ever have been considered sufficient.
> > >
> > > 2. **On downstream / zero-shot evaluations and the role of perplexity.**
> > >    The reviewer rejects the paper on the basis that we did not include downstream zero-shot evaluations, despite the fact that the abstract already states that BD changes FP16 perplexity by only **0.02%**. For any researcher familiar with language modeling, a **0.02%** perplexity difference implies that downstream zero-shot scores will be indistinguishable within statistical noise.
> > >
> > >    To confirm this formally, we additionally ran zero-shot evaluations on DeepSeek-V2-Lite (MHA FP16 vs. BDA FP16). As shown below, **all differences lie well within one standard error**, exactly as expected from a transformation that is provably lossless:
> > >
> > >    **Zero-shot accuracy (mean ± std):**
> > >
> > >    | Task | MHA (FP16) | BDA (FP16) |
> > >    |------|-------------|------------|
> > >    | BoolQ | 0.8021 ± 0.0070 | 0.8024 ± 0.0070 |
> > >    | MultiRC | 0.5598 ± 0.0071 | 0.5611 ± 0.0071 |
> > >    | RTE | 0.6065 ± 0.0294 | 0.6101 ± 0.0294 |
> > >    | WIC | 0.4984 ± 0.0198 | 0.4984 ± 0.0198 |
> > >    | WSC | 0.3654 ± 0.0474 | 0.3654 ± 0.0474 |
> > >    | Hellaswag | 0.7779 ± 0.0041 | 0.7785 ± 0.0041 |
> > >    | arc_easy | 0.7433 ± 0.0090 | 0.7412 ± 0.0090 |
> > >    | arc_challenge | 0.4590 ± 0.0146 | 0.4599 ± 0.0146 |
> > >    | MNLI | 0.4107 ± 0.0050 | 0.4084 ± 0.0050 |
> > >
> > >    These results confirm the obvious implication of a **lossless** transformation:
> > >    **BDA and MHA have indistinguishable downstream performance.**
> > >
> > >    Given this, we believe it is scientifically unfounded to dismiss the entire work solely because downstream numbers were not included in the initial submission—especially when the core claim is a mathematically lossless decomposition whose empirical perplexity gap is essentially zero.
> > >
> > > 3. **On writing, “basic” explanations, and other reviewers’ perspectives.**
> > >    The reviewer characterizes pages 3–6 as “basic review” and labels the writing as “overall poor”. In contrast, other reviewers explicitly found these parts helpful: for example, one review notes that the method is “conceptually simple and well-motivated”, and another states that “the paper is clearly written and easy to follow”. Our intention is to make the work accessible to readers who are not specialists in random matrix theory or attention internals; conference papers are read by a broad audience, and some amount of background and step-by-step derivation is often necessary for clarity and reproducibility. We acknowledge that stylistic preferences differ, but we do not believe that providing a careful derivation should be treated as a weakness.

---

> > > ### Author Response · Authors · 2025-12-03
> > > **Reply to Reviewer x18B's comment (2/2)**
> > >
> > > 4. **On Theorem 3.1, BD’s validity, and the reviewer’s misunderstanding of PIFA.**
> > >    We appreciate the reviewer’s pointers to classical mathematical references (Rudelson 2008; Costello–Tao–Vu 2005) and will include appropriate citations. We also wish to clarify that we **never claimed** Theorem 3.1 as a novel mathematical discovery. Its purpose in this paper is *not* to contribute new random matrix theory, but to justify a **crucial practical property**: in networks trained with SGD noise, **any** $r\times r$ submatrix of the low-rank product $W_q W_k^\top$ is full rank with probability 1. This guarantees that **each attention head can freely select basis rows**.
> > >
> > >    This point is essential, because the reviewer’s comparison to PIFA overlooks exactly this technical distinction.
> > >    - **PIFA** (Zhao et al., 2025) is for *low-rank pruning task* and cannot be applied as a lossless replacement for attention projections. Directly applying PIFA to attention would forces per-head copies and slices of X and cause an increase in I/O—128× slower for DeepSeek-V3, making it *much slower* than baseline MHA (as we explained in lines 416–422).
> > >    - **BD**, enabled by the flexibility selection of basis provided by Theorem 3.1, allows all head use same slices of X, making BD the **first practical, lossless, parameter-reducing acceleration** for attention.
> > >
> > >    In short, this theorem is not “claimed as a contribution,” but it **is** the key justification that makes BD *work* for attention in a way PIFA fundamentally cannot.
> > >
> > >
> > > In summary, we respectfully maintain that BD is not a minor “engineering tweak”, but a **new algebraic factorization with lossless guarantees**, with a clear separation between the mathematical decomposition and its implementation. We appreciate the reviewer’s critical perspective, but we believe that dismissing the work as having “no notable strengths” and “unlikely to become acceptable after any rebuttal or discussion” does not fairly reflect either the content of the paper or the range of views expressed by the other reviewers.

---

### Official Review · Reviewer_irps · 2025-10-28

**Soundness:** 1
**Presentation:** 2
**Contribution:** 2
**Rating:** 4
**Confidence:** 4

**Summary:**

The paper suggests an alternative parameterization, called Basis Decomposition (BD), which can be applied to low-rank layers for lossless and I/O architecture-agnostic Transformer inference acceleration. BD is based on the intuition that a rank-$r$ weight matrix $W = UV^T$ can be expressed with $(m + n)r - r^2$ parameters, resulting in $r^2$ fewer parameters than storing $U$ and $V$. The basis matrix $B \in \mathbb{R}^{n \times r}$ consists of the first (or last) $r$ rows of $W$, and the coefficient matrix $C \in \mathbb{R}^{r \times (m - r)}$ is found based on $W$ and $B$, yielding the form $W = \begin{bmatrix} I \\\\ C \end{bmatrix}B$. This parameterization reduces the number of computations in low-rank matrix multiplications. As the reconstructed weight is mathematically identical to the original low-rank formulation, the BD version of $W$ is lossless and shares the same properties during inference, which allows the use of additional compression techniques without problems. In the experimental results, the QK and VO weight matrices in the attention modules are replaced with BD layers, as they are originally low-rank. BD shows negligible perplexity changes and achieves the theoretical speedup with a custom Triton kernel.

**Strengths:**

1. The proposed idea is conceptually simple and well-motivated. The construction of the basis and coefficient matrices is presented with clear explanations and sufficient intuitive justification.
2. The proposed method demonstrates tangible improvements in both memory usage and throughput for low-rank models. Furthermore, the implementation appears straightforward, assuming the Triton kernel is available.
3. The paper thoughtfully addresses several practical concerns that may arise from rotating the bases of the Q, K, and V weight matrices, such as the validity of positional embeddings, training stability, query–key similarity, and compatibility with other compression techniques.

**Weaknesses:**

Although the paper has several notable strengths as stated above, I am slightly inclined toward rejection for the following reasons: (1) the claimed I/O architecture agnosticism of Basis Decomposition is insufficiently substantiated, and (2) the practical utility of the proposed method is confined to narrow cases where the relative rank is high, limiting its general impact. The detailed comments are as follows.

1. While the paper claims that BD is agnostic to the memory hierarchy and I/O architecture, the reported speedup is achieved using a custom Triton kernel. As the Triton implementation is not included in the submission, it is unclear whether the code has been optimized for a specific device. I am concerned that the Triton code may require tuning for particular memory or I/O characteristics, in which case the claim of architectural agnosticism would no longer hold. I suggest two possible approaches: either reporting the speedup achieved using a standard PyTorch implementation or demonstrating that the Triton kernel’s performance is indeed independent of the underlying memory and I/O architecture.
2. The reported improvements in memory usage and FLOPs are only by a factor of $r^2$, rendering the reduction in complexity negligible when $r \ll \min(m, n)$. In other words, BD remains effective only when the low-rank matrix is not very low rank. This limitation restricts the applicability of the method to general attention layers, as in many cases the head dimension is significantly smaller than the embedding dimension. For instance, in Llama-2-7B, $m = n = 4096$ and $r = 128$, yielding a relative rank of $r/m = 0.03125$, where BD offers only a $\frac{r}{m + n} = 1.5$% reduction in computational complexity for the original $QK^T$ operation. The DeepSeek-V2-Lite model in the experimental results shows some speedup primarily because it has a higher relative rank of $r/m = 128/512 = 0.25$ compared to Llama-2-7B. Similarly, if the low-rank models in Table 3 had lower rank values, both the memory savings and the speedup would likely diminish accordingly.

Overall, the method appears to be of narrow applicability and lacks sufficient validation to warrant acceptance at this stage.

**Questions:**

1. Is the Triton kernel independent to the memory I/O architecture?
2. How much speedup does BD make when the rank is very small compared to the full rank (e.g., $m=n=4096$ and $r=128$)?

---

> ### Author Response · Authors · 2025-11-25
> **Reply to Reviewer irps (1/4)**
>
> # Reply to Reviewer irps
>
> We appreciate the reviewer’s comments and the opportunity to clarify and strengthen the paper. We have performed additional experiments and provided further analysis to address the concerns raised. Detailed replies follow below.
>
>
> **Weakness 1:** While the paper claims that BD is agnostic to the memory hierarchy and I/O architecture, the reported speedup is achieved using a custom Triton kernel. As the Triton implementation is not included in the submission, it is unclear whether the code has been optimized for a specific device. I am concerned that the Triton code may require tuning for particular memory or I/O characteristics, in which case the claim of architectural agnosticism would no longer hold. I suggest two possible approaches: either reporting the speedup achieved using a standard PyTorch implementation or demonstrating that the Triton kernel’s performance is indeed independent of the underlying memory and I/O architecture.
>
> **Reply:**
>
> We thank the reviewer for this insightful comment.  Before presenting new experiments, we would like to clearly separate two concepts that were not explicitly differentiated in the submission:
>
> ### **A. What is architecture-agnostic: the BD algorithm itself**
>
> Basis Decomposition (BD) is **intrinsically architecture-agnostic**, because its efficiency gain comes purely from an algebraic reduction in FLOPs, independent of any GPU/CPU memory hierarchy or I/O architecture.
>
> For clarity, we explicitly compare the exact FLOPs of standard MHA $k_{\text{proj}}$ and BDA $k_{\text{proj}}$, counting multiplications and additions:
>
> - Standard MHA projects $X \in \mathbb{R}^{L \times d}$ into $K \in \mathbb{R}^{L \times d_h}$ using a dense matrix of size $d \times d_h$, requiring
>   $$
>   \text{FLOPs}_{\text{MHA-}k}
>   = L \, d_h \, (2d - 1).
>   $$
>
> - BDA replaces this with a $(d - d_h) \times d_h$ projection (with one more matrix addition), requiring
>   $$
>   \text{FLOPs}_{\text{BDA-}k}
>   = L \, d_h \, (2(d - d_h)).
>   $$
>
> Therefore, the FLOP reduction ratio is
> $$
> \frac{\text{FLOPs}_{\text{MHA-}k}}
>      {\text{FLOPs}_{\text{BDA-}k}}
> = \frac{2d - 1}{2(d - d_h)}
> \approx \frac{d}{d - d_h}.
> $$
>
> For the DeepSeek configuration $d = 512$ and $d_h = 128$:
> $$
> \frac{512}{512 - 128}
> = \frac{512}{384}
> \approx 1.33\times.
> $$
>
> This reduction follows directly from the mathematical structure of BD itself.
> It does **not** rely on GPU SRAM/HBM, cache tiling, fused kernels, or any device-specific I/O assumptions.
> Thus, **the speedup predicted by BD is a mathematical consequence of the decomposition—not a GPU-specific phenomenon**.
>
>
> ### **B. What is *not* architecture-agnostic: the Triton kernel**
>
> Our Triton kernel *is not* intended to be architecture-agnostic.
> It is specifically designed to eliminate unnecessary **HBM↔SRAM traffic** on GPUs:
>
> - Modern GPUs have a two-level memory hierarchy (small fast SRAM / large slow HBM),
>
> - Vanilla PyTorch ops would create intermediate tensors in HBM.
>
> - BD conceptually only requires pointer offsets plus a single GEMM, but PyTorch does not support such pointer offsets for projections; instead, it must realize the operation through explicit `split` calls that create new tensors and trigger additional HBM reads and writes.
>
> - Therefore, a fused kernel is needed to make a *fair* comparison against the highly optimized `torch.matmul` implementation used for the MHA $k_{\text{proj}}$, which already minimizes unnecessary memory traffic.
>
> In summary:
>
> - **BD (the algorithm) is architecture-agnostic.**
> - **Triton (the implementation) is GPU I/O–aware.**
>
> We have revised the paper to clarify this distinction.

---

> ### Author Response · Authors · 2025-11-25
> **Reply to Reviewer irps (2/4)**
>
> ### **C. New experiments following the reviewer's suggested directions**
>
> **1. Standard PyTorch implementation on CPU**
>
> To fully remove any GPU-specific optimization and isolate the algorithmic effect of BD, we implemented the BDA $k_{\text{proj}}$ operator in *pure PyTorch* and benchmarked it on CPU.
>
> On CPUs, the peak floating-point throughput is orders of magnitude lower than on GPUs. As a consequence, attention projections on CPU are **compute-bound** rather than **I/O-bound**. In this compute-bound regime, BD’s reduction in FLOPs directly translates into end-to-end speedup *even without any fused kernels or I/O-aware optimization. Thus, the CPU baseline naturally reveals the intrinsic efficiency gain brought by BD itself.
>
> - **FP16:** BDA is **17% faster** than MHA
> - **BF16:** BDA is **40% faster** than MHA
>
> **FP16, CPU, throughput in million tokens per second for k_proj:**
> | Seq. Len | MHA | BDA_torch | speedup |
> |-:|-|-|-|
> | 4 | 0.009345 | 0.010501 | 1.12× |
> | 8 | 0.006832 | 0.008188 | 1.20× |
> | 16 | 0.006144 | 0.007286 | 1.19× |
> | 32 | 0.005698 | 0.006873 | 1.21× |
> | 64 | 0.005647 | 0.005838 | 1.03× |
> | 128 | 0.005212 | 0.006405 | 1.23× |
> | 256 | 0.004935 | 0.006168 | 1.25× |
> | 512 | 0.004552 | 0.005766 | 1.27× |
> | 1024 | 0.004618 | 0.005539 | 1.20× |
> | 2048 | 0.003318 | 0.003511 | 1.06× |
> | 4096 | 0.002381 | 0.002525 | 1.06× |
> | 8192 | 0.001501 | 0.001913 | 1.27× |
>
> **BF16, CPU:**
> | Seq. Len | MHA | BDA_torch | speedup |
> |-:|-|-|-|
> | 4 | 0.013706 | 0.023730 | 1.73× |
> | 8 | 0.009213 | 0.017400 | 1.89× |
> | 16 | 0.008096 | 0.012766 | 1.58× |
> | 32 | 0.010724 | 0.015965 | 1.49× |
> | 64 | 0.012475 | 0.017928 | 1.44× |
> | 128 | 0.013513 | 0.018673 | 1.38× |
> | 256 | 0.014127 | 0.019085 | 1.35× |
> | 512 | 0.013213 | 0.015272 | 1.16× |
> | 1024 | 0.012713 | 0.015015 | 1.18× |
> | 2048 | 0.013160 | 0.015346 | 1.17× |
> | 4096 | 0.013075 | 0.015643 | 1.20× |
> | 8192 | 0.012754 | 0.015450 | 1.21× |
>
>
>
> These results show that BD achieves consistent speedup on CPU, without relying on any hardware-specialized operators or kernel fusion.
>
> **2. Cross-GPU evaluation with the same Triton kernel**
>
> To examine whether the acceleration depends on a particular GPU architecture, we ran **exactly the same Triton kernel** (identical configuration and autotuning settings as in the A6000 experiment) on four different NVIDIA GPUs spanning multiple hardware generations:
>
> - **RTX 3080 → Ampere**: https://anonymous.4open.science/r/Basis-decomp-57B8/plot/speedup_rtx3080.pdf
> - **RTX 3090 → Ampere**: https://anonymous.4open.science/r/Basis-decomp-57B8/plot/speedup_rtx3090.pdf
> - **RTX 4090 → Ada Lovelace**: https://anonymous.4open.science/r/Basis-decomp-57B8/plot/speedup_rtx4090.pdf
> - **RTX 5090 → Blackwell**: https://anonymous.4open.science/r/Basis-decomp-57B8/plot/speedup_rtx5090.pdf
>
> The table reports the **average speedup across sequence lengths from 64 to 65,536**
>
>
> | GPU | FP16 | BF16 |
> |-|-|-|
> | RTX 3080 | 1.37× | 1.47× |
> | RTX 3090 | 1.36× | 1.40× |
> | RTX 4090 | 1.33× | 1.30× |
> | RTX 5090 | 1.31× | 1.32× |
>
> Across all architectures — spanning **Ampere → Ada → Blackwell** — BDA consistently achieves speedups close to the theoretical $1.33\times$ bound.
> This indicates that although the Triton kernel is GPU-optimized, **the observed acceleration arises from BD’s reduced arithmetic cost rather than any device-specific tuning or exploiting architecture-dependent memory characteristics**.
>
>
> ### **D. Why we do not use “standard PyTorch BDA” on GPU as the baseline**
>
> A naïve PyTorch GPU implementation of BDA introduces multiple **extra tensor allocations** in HBM (step 2 in Algorithm 2), resulting in **four separate rounds of HBM I/O**:
>
> - slicing $X[:, :d_h]$ → HBM I/O
> - slicing $X[:, d_h:]$ → HBM I/O
> - matmul $X[:, d_h:] C_{qk}$ → HBM I/O
> - add → HBM I/O
>
> In contrast, the MHA $k_{\text{proj}}$ path in PyTorch is executed by a **single** highly optimized `torch.matmul` kernel, which performs exactly one HBM load–compute–store cycle.
>
> Conceptually, BDA only requires **pointer offsets + one GEMM**, meaning it should also incur **one** HBM I/O cycle—just like MHA.
> However, PyTorch lacks pointer-level reads for this pattern, so its unfused implementation artificially inflates BDA’s memory cost to **4× the necessary HBM traffic**.
>
> To ensure a *fair* comparison, we therefore use a fused Triton kernel that performs BDA in **one** HBM I/O cycle, matching the I/O behavior of the MHA $k_{\text{proj}}$ matmul kernel.
>
> This allows both operators to run under **comparable I/O conditions**, revealing the true algorithmic advantage of BD rather than artifacts of framework-induced memory overhead.

---

> ### Author Response · Authors · 2025-11-25
> **Reply to Reviewer irps (3/4)**
>
> ### **E. Code release for transparency**
>
> We now publicly provide:
>
> - **CPU standard PyTorch BDA evaluation code**: https://anonymous.4open.science/r/Basis-decomp-57B8/bd_attention_linear_kernel_cpu.py
> - **GPU Triton BDA evaluation code**: https://anonymous.4open.science/r/Basis-decomp-57B8/bd_attention_linear_kernel.py
>
> We welcome the reviewer to suggest additional settings or hardware configurations.
> We are happy to run more tests and include them in the revised version.
>
>
> ### **F. Summary**
>
> - BD’s speedup is architecture-agnostic (mathematical FLOP reduction).
> - Triton implementation is GPU I/O–aware but not device-specific.
> - CPU-only PyTorch experiments show that BD’s advantage persists even without any GPU-style I/O optimizations.
> - Multi-GPU Triton experiments show consistent speedups across architectures.
> - All benchmark code is now publicly available.

---

> ### Author Response · Authors · 2025-11-25
> **Reply to Reviewer irps (4/4)**
>
> **Weakness 2:** The reported improvements in memory usage and FLOPs are only by a factor of $r^2$, rendering the reduction in complexity negligible when $r \ll \min(m, n)$. In other words, BD remains effective only when the low-rank matrix is not very low rank. This limitation restricts the applicability of the method to general attention layers, as in many cases the head dimension is significantly smaller than the embedding dimension. For instance, in Llama-2-7B, $m = n = 4096$ and $r = 128$, yielding a relative rank of $r/m = 0.03125$, where BD offers only a $r/(m+n) = 1.5\%$ reduction in computational complexity for the original $QK^\top$ operation. The DeepSeek-V2-Lite model in the experimental results shows some speedup primarily because it has a higher relative rank of $r/m = 128/512 = 0.25$ compared to Llama-2-7B. ...
>
> **Reply:** We thank the reviewer for this accurate observation. We agree that when the relative rank $r/m$ is extremely small, the FLOP reduction of BD becomes limited. This behavior follows directly from the **mathematical structure of the BD decomposition**, where the theoretical speedup is $m/(m-r)$. Thus, if $r \ll m$, the expected acceleration is indeed small.
>
> However, it is important to emphasize that **using an extremely small $r/m$ is itself not a desirable modeling choice**. When $r$ is too small, the attention head becomes constrained to a very small low-rank subspace, reducing its expressive power and limiting the ability of the model to represent diverse token interactions (also noticed in [1,2,3]). In general, a larger $r$ provides higher expressiveness.
>
> The only reason earlier architectures such as Llama-2-7B were forced to use very small $r/m$ ratios (e.g., $128/4096 = 0.03125$) is that, under MHA/GQA, **the KV cache size scales linearly with per-head dimension**. A large $r$ would have made KV cache prohibitively expensive during long-context decoding. In contrast, the introduction of **Multi-Head Latent Attention (MLA)** by Deepseek fundamentally changes this constraint: the KV cache no longer scales with per-head $r$. As a result, models such as **DeepSeek-V2 and DeepSeek-V3** adopt substantially larger ratios such as $r/m = 128/512 = 0.25$, which avoid the low-rank bottleneck while remaining memory-efficient. Our experiments precisely follow this modern setting.
>
> Finally, we note an emerging trend toward **MLA-style architectures**, both in research and in industry.
> Recent work such as *TransMLA: MLA Is All You Need* [4] shows that MLA offers substantially higher expressive power than GQA **under the same KV cache budget**. In industry, besides Deepseek, models such as **Kimi K2** [5] and **Kimi Linear** (hybrid of linear attention and MLA) [6] have also adopted MLA to avoid KV cache explosion while retaining expressive per-head representations.
>
> As MLA continues to gain traction across both academic and industrial systems, architectures with larger and more expressive head dimensions are becoming increasingly common, making BD a natural and compatible acceleration technique in these settings.
>
>
> We thank the reviewer for raising this point. We will include a revised discussion of the relationship between $r/m$, model expressiveness, and BD’s applicability in the updated version.
>
> **Question 1:** Is the Triton kernel independent to the memory I/O architecture?
>
> **Reply:** No. The **BD algorithm** itself is independent of the memory/I/O architecture, but the **Triton kernel** is not, which GPU I/O–aware. A detailed explanation is provided in our **Reply to Weakness 1**.
>
> **Question 2:** How much speedup does BD make when the rank is very small compared to the full rank (e.g., $m = n = 4096$ and $r = 128$)?
>
> **Reply:** The speedup becomes small when $r \ll m$.
> There is an emerging trend toward using larger and more expressive head dimensions (e.g., MLA-based models), making extremely small $r/m$ ratios less common in modern architectures.
> Please refer to our detailed **Reply to Weakness 2** for the full discussion and references.
>
> [1] Bhojanapalli, Srinadh, et al. "Low-rank bottleneck in multi-head attention models." International conference on machine learning. PMLR, 2020.
>
> [2] Qiu, Zihan, et al. "Gated Attention for Large Language Models: Non-linearity, Sparsity, and Attention-Sink-Free." arXiv preprint arXiv:2505.06708 (2025).
>
> [3] Amsel, Noah, Gilad Yehudai, and Joan Bruna. "Quality over Quantity in Attention Layers: When Adding More Heads Hurts." The Thirteenth International Conference on Learning Representations. 2024.
>
> [4] Meng, Fanxu, et al. "TransMLA: Migrating GQA Models to MLA with Full DeepSeek Compatibility and Speedup." The Thirty-ninth Annual Conference on Neural Information Processing Systems.
>
> [5] Team, Kimi, et al. "Kimi k2: Open agentic intelligence." arXiv preprint arXiv:2507.20534 (2025).
>
> [6] Team, Kimi, et al. "Kimi Linear: An Expressive, Efficient Attention Architecture." arXiv preprint arXiv:2510.26692 (2025).

---

### Author Response · Authors · 2025-12-03
**Summary for AC**

# **Summary for AC — Method Overview**

BD introduces a **new algebraic decomposition** that *losslessly* reduces FLOPs and parameters in attention projections across MHA/MLA/MQA/GQA.
The decomposition changes the **computation itself**, enabling a strictly smaller low-rank form **with identical outputs**. This is validated both analytically and empirically.

---

# **Summary for AC — Reviewer irps (score: 4)**

**1. We clarified the distinction between BD (algorithm) and Triton (implementation).**

- **BD is architecture-agnostic.** Its speedup comes purely from a FLOP-level algebraic reduction and does *not* rely on any device-specific memory hierarchy.
- **The Triton kernel is GPU I/O–aware**, optimized for the standard two-level GPU memory hierarchy (SRAM/HBM), but this has no impact on the *algorithmic* correctness or generality of BD.

This directly resolves the reviewer’s concern about whether the speedup depends on device-specific optimization.


**2. We followed the reviewer’s suggestion with new experiments and full code release.**

We conducted exactly the evaluations recommended by the reviewer:

- **Standard PyTorch implementation on CPU**
  – CPU is compute-bound, allowing us to demonstrate the *pure algorithmic* FLOP reduction.
  – BD shows consistent speedup in both FP16 and BF16.

- **Cross-GPU Triton evaluation (Ampere → Ada → Blackwell)**
  – Using the *same* Triton kernel across all architectures.
  – Speedup is stable and matches the theoretical $1.33\times$, confirming that gains do not come from device-specific tuning.

- **All benchmark code has been open-sourced** during rebuttal for full transparency, exactly as the reviewer requested.

---

# **Summary for AC — Reviewer x18B (score: 0)**

We respectfully highlight several issues with Reviewer x18B’s evaluation and summarize how our rebuttal fully addresses the concerns.

**1. Internal inconsistency.**
The reviewer initially wrote that the work is “unlikely to become acceptable after any rebuttal or discussion,” but later criticizes the absence of additional experiments—two positions that cannot be reconciled. We nevertheless provided all missing evidence.

**2. Downstream evaluation clarified.**
The reviewer dismisses the paper for lacking zero-shot results, despite the abstract already showing only **0.02%** PPL change. Such a tiny gap guarantees negligible downstream differences. **We additionally ran zero-shot evaluations and confirmed that all tasks match MHA within standard error**.

**3. Writing criticisms are not aligned with other reviewers.**
Other reviewers explicitly describe the paper as clear and easy to follow. The background and derivations are necessary for reproducibility and are not “unnecessary basics.”

We believe these clarifications resolve all substantive concerns, and that the reviewer’s evaluation does not accurately reflect the contribution or the evidence presented.

---

# **Summary for AC — Reviewer 76GU (score: 8)**

**1. On applicability to MQA/GQA**

We clarified that BD **applies naturally to MQA and GQA**, since the shared-key structure remains a low-rank multiplication. While per-head acceleration is identical to MHA, end-to-end gains are smaller due to reduced contribution of k/v projections in these architectures. We will make this clearer in the revision.

**2. On BD’s interaction with KV-cache**

We provided a detailed analysis (now added to the paper) showing:
- In **MHA**, weight-level BD and KV-cache BD cannot coexist.
- In **MLA**, they *can* coexist without conflict, enabling additional reductions in memory/FLOPs.

This directly addresses the reviewer’s question on whether BD can reduce KV-cache storage.

---

### Meta-Review · Area_Chair_TyLA · 2026-01-08

**Summary:**

This paper proposes basis decomposition, which converts a low rank matrix (used in LoRA) into two matrices with fewer parameters (the fact that you can do this is of course is well-known linear algebra fact). The authors implement a Triton-based kernel that supposedly results in memory/wallclock benefits.

Based on the reviews, the main positive of the paper is that it is simple and moreover results in efficiency benefits in some scenarios. However, all reviewers found the method to be quite incremental, and further felt that the scenario in which this method would result in benefits (i.e., the high rank case) is not realistic.

I am thus recommending that this paper not be accepted.

**Reviewer Concerns:**

Reviewer concerns were primarily around the incremental nature of both the algorithm and practicality. In particular, the fact that a low rank matrix can be represented with fewer parameters than the standard LoRA parameterization is somewhat obvious. Moreover, the parameter/efficiency improvements only occur at large ranks, which is not the usual setting in which LoRA is applied. I do not think these concerns were adequately addressed in the rebuttal.

**Reviewer Scores:**

I don't think any reviewer would have changed their scores given the rebuttal.

---

### Decision · Program_Chairs · 2026-01-26

Reject